# Prompt Perturbation for Reliable LLM Evaluation over Comparison Graphs

Dong Huang

Department of Statistics and Data Science, Tsinghua University

Jianbo Sun

Department of Statistics and Data Science, Tsinghua University

Pengkun Yang*

Department of Statistics and Data Science, Tsinghua University

**Abstract**

Evaluating large language models (LLMs) is important for understanding their capabilities, comparing competing systems, and supporting the deployment of reliable models in practice. For open-ended tasks, pairwise evaluation has become a popular paradigm, in which two responses to the same prompt are compared and the resulting judgments are aggregated into an overall ranking. A central challenge of this paradigm is intransitivity: the induced comparison outcomes may fail to support any coherent global ranking. For example, one may observe cyclic preferences such as $A \succ B \succ C \succ A$, or inconsistencies involving ties such as $A \equiv B \equiv C \not\equiv A$. Such contradictions make the resulting leaderboard unstable and challenging to interpret. In this paper, we propose a prompt perturbation framework for improving the consistency of pairwise LLM evaluation. Our approach generates perturbed variants of each prompt, uses the resulting comparison graphs to identify and filter out structurally inconsistent comparison patterns, and then applies standard ranking methods to the filtered comparisons. A key feature of the proposed framework is that graph-level structural consistency is incorporated explicitly into the evaluation pipeline before ranking aggregation. This provides a simple and principled way to reduce cyclic inconsistencies and improve the reliability of LLM rankings.

**Keywords:** LLM-as-a-judge, prompt perturbation, ranking, intransitivity

**Mathematics Subject Classification (2020):** 62F03, 68T50

## 1   Introduction

The rapid progress of large language models (LLMs) as general-purpose systems for complex tasks (OpenAI et al., 2023; Touvron et al., 2023; Guo et al., 2025) raises an important challenge: how to establish a gold standard for LLM evaluation. The standard pipeline for evaluating

---

*Corresponding author: yangpengkun@tsinghua.edu.cn

LLMs relies on benchmarks (Hendrycks et al., 2021; Cobbe et al., 2021; Srivastava et al., 2023) that assess model performance across a collection of representative tasks and aggregate the results into summary scores. Although many existing benchmarks have been proposed for LLM evaluation, they often fail to capture the full range of model capabilities, especially in open-ended and domain-specific settings. In many practical scenarios, the goal of LLM evaluation is not merely to produce a universal ranking over all models, but rather to identify the most suitable model for a specific domain or application.

To overcome the limitations on LLM evaluation, an increasingly common strategy is to evaluate LLMs through pairwise comparisons (Zheng et al., 2023; Chiang et al., 2024; Dubois et al., 2024), where two responses to the same prompt are judged relative to each other. Such judgments may be provided either by human annotators or by a strong judge model. Although human evaluation remains the most direct way to assess open-ended generation, it is costly and challenging to scale. This has led to the widespread adoption of LLM-as-a-judge methods, in which a powerful model automatically compares candidate outputs. Empirical evidence suggests that such judge models can align well with human preferences; for example, Zheng et al. (2023) reported that GPT-4 achieves agreement with human judgments at a level comparable to human annotators themselves. In many existing pairwise evaluation pipelines, each target model is compared only against a fixed reference model (Li et al., 2023; Dubois et al., 2024), which yields an efficient but potentially unstable ranking procedure. In particular, rankings may depend heavily on the choice of the reference. As a result, more recent frameworks increasingly rely on broader pairwise comparisons among many models and then aggregate these comparison outcomes into an overall ranking. A convenient way to view modern pairwise LLM evaluation is as a four-stage pipeline, in which pairwise judgments are first collected and then aggregated through an induced comparison graph. Traditional pairwise LLM evaluation typically consists of the following four steps:

1. *Input.* A collection of evaluation inputs such as prompts.

2. *LLM-as-a-judge.* For each input, two candidate responses are compared by a judge model, which returns a pairwise preference or a tie.

3. *Comparison graph.* These pairwise outcomes are then aggregated into a comparison graph over models.

4. *Ranking method.* A ranking procedure is finally applied to this graph to produce an overall ordering of the models.

Among these four components, the comparison graph plays a central role, since it serves as the intermediate object through which local pairwise judgments are translated into a global ranking.

Recent work has shown that pairwise LLM evaluation can exhibit substantial intransitivity (Xu et al., 2025; Wang et al., 2026), so the induced comparison graph may fail to admit any coherent global ranking. In particular, the comparison outcomes may contain both cycles ($A \succ B \succ C \succ A$) and equivalence contradictions ($A \equiv B \equiv C \not\equiv A$), making the resulting leaderboard unstable. Indeed, the prevalence of cycles in the comparison graph is a key indicator of ranking reliability (Kenyon-Mathieu and Schudy, 2007), as a larger fraction of cycles suggests greater inconsistency with any global ordering. This raises a natural question:

*How can we design a pairwise evaluation pipeline that improves the structural consistency of the induced comparison graph?*

In this paper, we propose a prompt perturbation framework for reducing cyclic inconsistencies in pairwise LLM evaluation. Specifically, for each input prompt, we construct a collection of perturbed prompts and use them to obtain a richer set of pairwise judgments among candidate models. These judgments induce a family of comparison graphs, one for each perturbed prompt. We then evaluate the structural consistency of each graph through its cycle ratio, and retain only those graphs whose cycle ratio is below a prescribed threshold. By filtering out highly inconsistent comparison graphs prior to aggregation, the proposed procedure improves the global consistency of the resulting ranking. Finally, based on the retained comparison outcomes, we perform ranking aggregation using standard models for pairwise comparison data, such as the Bradley–Terry model (Bradley and Terry, 1952) and the Davidson model (Davidson, 1970). Specifically, for each prompt $x_i$, we generate $m$ perturbed variants $x_{i,1}, \cdots, x_{i,m}$. Each of the original and perturbed prompts then induces a comparison graph, denoted by $G_i, G_{i,1}, \cdots, G_{i,m}$. A key feature of our method is the explicit use of structural consistency at the comparison graph level as a criterion for filtering before ranking aggregation. This filtering step is intended to discard prompt instances that induce highly unstable local preference structures before downstream ranking aggregation.

Indeed, prompt perturbation is a widely used methodology for probing and improving the robustness of LLM systems. The basic idea is to replace a given input prompt by a collection of semantically similar variants, such as paraphrases (Sun et al., 2023), reordered instructions (Pezeshkpour and Hruschka, 2024), or other meaning-preserving perturbations (Qiang et al., 2024), and then examine the stability of the resulting outputs. This paradigm has been extensively adopted in recent studies on prompt robustness and evaluation, which show that LLM behavior can vary substantially under seemingly minor prompt changes, and that perturbation-based procedures provide an effective way to expose and control such instability (Zhu et al., 2024; Qiang et al., 2024).

## 1.1 Related Work

**LLM-as-a-judge**   The rapid progress of large language models (LLMs) (OpenAI et al., 2023) has driven remarkable advances across a wide range of applications, which in turn has motivated the development of the "LLM-as-a-judge" paradigm (Wang et al., 2023; Liu et al., 2023). In this paradigm, LLMs are used to evaluate model outputs by assigning scores, producing rankings, or selecting preferred responses. More recently, dedicated judge models have also been developed to support this paradigm, including fine-tuned or specialized evaluators such as JudgeLM and Prometheus (Zhu et al., 2025; Kim et al., 2024). Beyond evaluation, LLM-as-a-judge has also been adopted as a scalable source of supervision in other stages of LLM development, including alignment (Lee et al., 2024), retrieval (Upadhyay et al., 2025), and reasoning (Chen et al., 2025). These developments highlight the growing role of LLM judges not only as evaluation tools, but also as general-purpose components in the broader LLM pipeline.

**Intransitivity in LLM Ranking.** Recent work has shown that pairwise LLM evaluation may exhibit substantial intransitivity, so there may be no coherent global ranking consistent with the induced comparison graph. In particular, Xu et al. (2025) showed that intransitive preferences in AlpacaEval (Dubois et al., 2024) make system rankings sensitive to the choice of the reference baseline, and advocated broader pairwise comparison designs with Bradley–Terry aggregation (Bradley and Terry, 1952). Furthermore, Wang et al. (2026) identified systematic inconsistencies in LLM-as-a-judge pipelines, including cyclic preferences and contradictions involving ties. More broadly, recent work on LLM ranking has emphasized the importance of robust comparison design and aggregation for obtaining reliable leaderboards (Daynauth et al., 2025; Gao et al., 2025). Also, Liu et al. (2025) show that the absence of Condorcet cycles is necessary and sufficient for representing preferences by a scalar reward model, providing a complementary theoretical perspective on why cyclic inconsistencies are problematic for score-based aggregation. These findings suggest that intransitivity is a central challenge in pairwise LLM ranking and motivate methods that explicitly improve the consistency of the induced comparison graph.

**Ranking Methods** Classic methods for ranking from pairwise comparison graphs can be divided into three categories. The simplest approach is rank-by-wins (Copeland-style scoring) (Coleman, 1966), which orders items according to the number of pairwise wins. A more principled line of work assigns latent scores to items based on pairwise outcomes. A widely used example is the Elo rating system (Elo, 1978), which updates scores sequentially after each comparison, while the Bradley–Terry model (Bradley and Terry, 1952) provides the standard probabilistic formulation for paired comparisons and the Davidson model (Davidson, 1970) further extends Bradley–Terry to allow ties. A complementary graph-based approach is Rank Centrality (Negahban et al., 2017), which constructs a Markov chain from the comparison graph and ranks items via its stationary distribution; conceptually, it is closely related to PageRank (Page et al., 1999), and its statistical guarantees depend explicitly on graph properties such as connectivity and spectral structure.

In LLM evaluation, these classical ranking methods are primarily used as aggregation tools for pairwise judgments. Recent work compared ranking algorithms such as Elo, Bradley–Terry, Glicko, and Markov-chain methods in head-to-head LLM evaluation, emphasizing their differences in transitivity, stability, and sensitivity (Daynauth et al., 2025). In addition, Gao et al. (2025) decomposed automatic LLM system ranking into several pipeline components and showed that the final leaderboard depends not only on the judge model, but also on the aggregation method itself. This line of work mainly studies how to aggregate a given set of pairwise outcomes, whereas our focus is on improving the comparison graph itself before aggregation by reducing cyclic inconsistency.

**Prompt Perturbation.** Prompt perturbation has been widely used to study and improve the robustness of LLM systems. Recent work has shown that LLM behavior can vary substantially even under minor prompt changes, making prompt perturbation a useful tool for exposing sensitivity and improving robustness (Qiang et al., 2024; Agrawal et al., 2025). Relatedly, perturbation-based analysis has also been applied to LLM-based evaluators, showing

that evaluation pipelines themselves can be sensitive to prompt variations and other input per-turbations (Chaudhary et al., 2024). In contrast to this line of work, which mainly studies robustness at the response or evaluator level, our focus is on how prompt perturbation can be used to improve the structural consistency of the induced comparison graph in pairwise LLM evaluation.

## 1.2 Our Contribution

We study the problem of ranking large language models (LLMs) from pairwise evaluation data. A central challenge in this setting is that the induced comparison graph may exhibit substantial intransitivity, including both cyclic preferences and inconsistencies involving ties, which makes the resulting leaderboard unstable. To address this issue, we propose a prompt perturbation framework for improving the global consistency of pairwise LLM evaluation.

Our framework enriches the evaluation process by constructing multiple perturbed variants of each input prompt and collecting pairwise comparison outcomes under these perturbed prompts. Each perturbed prompt induces a comparison graph over the candidate models. We quantify the structural consistency of these graphs through their cycle ratios, and retain only those graphs whose cycle ratios fall below a prescribed threshold. Based on the retained comparison outcomes, we then perform ranking aggregation using standard models for pairwise comparison data, such as the Bradley–Terry model and the Davidson model.

The main contribution of this work is to incorporate graph-level structural consistency ex-plicitly into the LLM evaluation pipeline before ranking aggregation; see Figure 1 for an il-lustration. This yields a simple and principled procedure for filtering out highly inconsistent local comparison structures and reducing cyclic inconsistencies in the induced comparison graph. More broadly, our work highlights a graph-based perspective on LLM evaluation, in which the quality of a ranking depends not only on how pairwise judgments are aggregated, but also on how the underlying comparison graph is constructed.

# 2 Statistical Models

## 2.1 Problem Formulation

We consider the problem of ranking a collection of large language models (LLMs) based on pairwise comparisons. Let $\mathcal{M} = \{M_1, M_2, \cdots, M_n\}$ denote the set of candidate models and $\mathcal{X} = \{x_1, x_2, \cdots, x_t\}$ denote a collection of evaluation prompts from different task domains. For each $x_i \in \mathcal{X}$, we compare the responses produced by any two models $M_j, M_{j'} \in \mathcal{M}$ across all pairs $1 \le j < j' \le n$ through a judge model and record the comparison outcome. We then obtain a directed comparison graph $G_i$ with vertex set $V(G_i)$ and edge set $E(G_i)$ defined as

$$V(G_i) = \{1, 2, \cdots, n\}, \quad E(G_i) = \{(j, j') : j \succ j' \text{ under prompt } x_i\}.$$

Here, we write $j \succ j'$ if the judge prefers the response of $M_j$ to that of $M_{j'}$. We also allow both $(j, j')$ and $(j', j)$ to be present in $E(G_i)$, corresponding to the case where the comparison between $M_j$ and $M_{j'}$ results in a tie. In this case, we write $j \sim j'$. In the subsequent cycle-

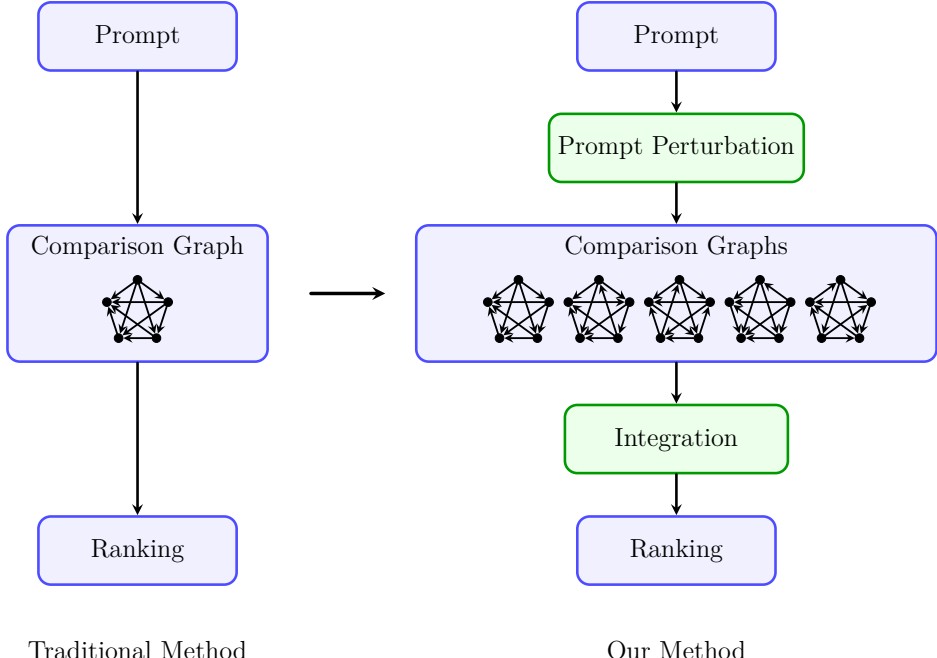

Figure 1: Comparison between the traditional pipeline and our prompt-perturbation-based aggregation framework.

counting step, such ties are represented by reciprocal directed edges, so they are included in the enumeration of directed triangles and quadrilaterals; cycles formed purely by mutual ties are later subtracted from the bad-cycle count. See Section 3.1 for further details. In Section 2.2, we first consider the Bradley–Terry model, where ties are not allowed and each comparison yields a strict preference between the two models. We then introduce the Davidson model, which extends this framework by explicitly allowing ties in pairwise comparisons. In practice, ties are sometimes handled heuristically by assigning each model 0.5 win and 0.5 loss; see Section 3 for further details.

Let $\mathbf{W} \in \mathbb{R}^{n \times n}$, where $\mathbf{W}_{ij}$ denotes the number of times that model $M_i$ is preferred to model $M_j$ over all pairwise comparisons, for each $1 \le i \ne j \le n$. Similarly, let $\mathbf{T} \in \mathbb{R}^{n \times n}$, where $\mathbf{T}_{ij}$ denotes the number of ties between models $M_i$ and $M_j$. These quantities are aggregated from the collection of comparison graphs $\mathcal{G} = (G_1, G_2, \cdots, G_t)$, where each $G_i$ corresponds to the comparison graph induced by prompt $x_i$. We assume that there exists a latent score vector $\theta = (\theta_1, \theta_2, \cdots, \theta_n)$, which determines the pairwise comparison probabilities through $\mathbb{P}_{i \succ j}(\theta_i, \theta_j)$, $\mathbb{P}_{i \prec j}(\theta_i, \theta_j)$, and $\mathbb{P}_{i \sim j}(\theta_i, \theta_j)$. For each pair of models $(M_i, M_j)$, the observed comparison outcomes across prompts fall into three categories: $M_i \succ M_j$, $M_i \prec M_j$, and $M_i \sim M_j$. Accordingly, the likelihood function is defined as

$$\mathcal{L}(\theta; \mathcal{G}, \mathbf{W}, \mathbf{T}) = \prod_{1 \le i < j \le n} \frac{N_{ij}!}{\mathbf{W}_{ij}! \, \mathbf{W}_{ji}! \, \mathbf{T}_{ij}!} \mathbb{P}_{i \succ j}(\theta_i, \theta_j)^{\mathbf{W}_{ij}} \mathbb{P}_{i \prec j}(\theta_i, \theta_j)^{\mathbf{W}_{ji}} \mathbb{P}_{i \sim j}(\theta_i, \theta_j)^{\mathbf{T}_{ij}}, \quad (1)$$

where $N_{ij} = \mathbf{W}_{ij} + \mathbf{W}_{ji} + \mathbf{T}_{ij}$ is the total number of comparisons between models $M_i$ and $M_j$. In the no-tie setting, we have $\mathbf{T}_{ij} = 0$ for all $1 \le i, j \le n$. The estimator $\hat{\theta}$ is then obtained by maximizing the log-likelihood function $\ell(\theta) = \log \mathcal{L}(\theta; \mathcal{G}, \mathbf{W}, \mathbf{T})$.

## 2.2 Ranking Models

The most widely used probabilistic model for pairwise comparison was first proposed by Zermelo (1929) and later rediscovered by Bradley and Terry (1952). This model assumes the existence of a latent score vector $\theta = (\theta_1, \theta_2, \cdots, \theta_n)$, and specifies the win and loss probabilities as

$$\mathbb{P}_{i \succ j}(\theta_i, \theta_j) = \frac{e^{\theta_i}}{e^{\theta_i} + e^{\theta_j}}, \quad \mathbb{P}_{i \prec j}(\theta_i, \theta_j) = \frac{e^{\theta_j}}{e^{\theta_i} + e^{\theta_j}}. \tag{2}$$

Consequently, the log-likelihood function $\ell(\theta)$ is given, up to an additive constant, by

$$\ell(\theta) \propto \sum_{1 \leq i < j \leq n} \left[ -\mathbf{W}_{ij} \log(1 + e^{\theta_j - \theta_i}) - \mathbf{W}_{ji} \log(1 + e^{\theta_i - \theta_j}) \right]. \tag{3}$$

The maximum likelihood estimator is obtained by maximizing $\ell(\theta)$ subject to an identifiability constraint, for example $\sum_{i=1}^n \theta_i = 0$ or $\theta_1 = 0$, since only pairwise score differences are identifiable in the Bradley–Terry model. The objective function is concave in $\theta$, so the estimation problem can be solved efficiently by standard convex optimization methods. In practice, the estimator can be computed by standard iterative methods such as Newton's method (Turner and Firth, 2012), fixed-point iteration (Yan et al., 2015), or minorization–maximization algorithms (Hunter, 2004). After obtaining the estimator $\widehat{\theta}$, a global ranking of the $n$ items is induced by ordering the estimated scores.

Note that the Bradley–Terry model does not allow ties, where $\mathbb{P}_{i \sim j}(\theta_i, \theta_j) \equiv 0$ for all $1 \leq i < j \leq n$. Therefore, it is not well suited to applications where tied outcomes may occur with non-negligible probability such as LLM evaluation. Although one common heuristic is to treat each tie as half a win and half a loss for each model (see, e.g., Chiang et al. (2024)), it is more natural to consider probabilistic models that explicitly incorporate ties.

We now introduce the Davidson model (Davidson, 1970), which extends the Bradley–Terry model by incorporating an additional parameter $\nu$ to account for ties:

$$\mathbb{P}_{i \succ j}(\theta_i, \theta_j) = \frac{e^{\theta_i}}{e^{\theta_i} + e^{\theta_j} + \nu \sqrt{e^{\theta_i} e^{\theta_j}}},$$

$$\mathbb{P}_{i \prec j}(\theta_i, \theta_j) = \frac{e^{\theta_j}}{e^{\theta_i} + e^{\theta_j} + \nu \sqrt{e^{\theta_i} e^{\theta_j}}},$$

$$\mathbb{P}_{i \sim j}(\theta_i, \theta_j) = \frac{\nu \sqrt{e^{\theta_i} e^{\theta_j}}}{e^{\theta_i} + e^{\theta_j} + \nu \sqrt{e^{\theta_i} e^{\theta_j}}},$$

where $\nu \geq 0$. In particular, when $\nu = 0$, the Davidson model reduces to the Bradley–Terry model. The parameter $\nu$ controls the relative likelihood of tied outcomes, with larger $\nu$ corresponding to a higher probability of tied outcomes. The log-likelihood function is obtained by substituting the above probabilities into (1). The maximum likelihood estimator is then defined as the solution to the resulting optimization problem under the constraint $\nu \geq 0$, and can be computed by standard optimization methods.

However, the Bradley–Terry and Davidson models are fundamentally score-based and hence transitivity-oriented aggregation models. This is consistent with the perspective of Liu et al. (2025), who show that scalar reward representations are possible exactly when the preference

relation is free of Condorcet cycles. When a comparison graph contains many short directed cycles, the data exhibit a large cyclic component that cannot be well explained by any single global score vector. In such cases, directly fitting a Bradley–Terry or Davidson model forces incompatible local preferences into a transitive parametric form, which may lead to unstable or distorted rankings (Spearing et al., 2023). This motivates us to adopt a truncation method on comparison graphs with large cycle counts; see Section 3.1 for further details.

# 3 Methods

## 3.1 Prompt Perturbation and Graph Truncation

Prompt perturbation is a commonly used strategy for improving the robustness of LLM evaluation, since judgments obtained from a single prompt template can be sensitive to superficial wording differences even when the underlying evaluation objective remains unchanged. Recent work has shown that single prompt evaluation can be brittle, and that more reliable conclusions can often be obtained by comparing model behavior across multiple semantically equivalent prompt formulations (Mizrahi et al., 2024; Maia Polo et al., 2024; Chaudhary et al., 2024). The underlying idea is to replace one prompt template with several semantic-preserving rephrasings, thereby reducing prompt-specific noise in the final judgment.

For each original comparison prompt, we generate multiple candidate perturbations, retain those that preserve the semantic meaning of the original task, and use them to obtain more stable pairwise judgments for the subsequent construction of the comparison graph. Specifically, for any original prompt $x_i \in \mathcal{X}$ with $1 \leq i \leq t$, let $x_{i,1}, \ldots, x_{i,m}$ denote the perturbed prompts generated from $x_i$. In our implementation, the perturbations are generated by GPT-5.2 (OpenAI, 2025); see Section 4.1 for details. Ideally, the generation prompt should request semantically equivalent rephrasings that preserve the original intent, constraints, entities, and requirements while changing only the surface representation. For each $1 \leq j \leq m$, let $G_{i,j}$ be the comparison graph induced by prompt $x_{i,j}$. Without loss of generality, we further define $x_{i,0} = x_i$ and $G_{i,0} = G_i$. The resulting collection of comparison graphs contains richer information than a single graph, and we will leverage this additional information to perform a more robust ranking.

We then introduce our integration methods based on the comparison graphs $\{G_{i,j} : 1 \leq i \leq t,\ 0 \leq j \leq m\}$. Recent studies suggest that inconsistency is closely related to the reliability of LLM evaluation (Xu et al., 2025; Wang et al., 2026). A natural graph-based measure of such inconsistency is the abundance of cycles in the comparison graph. Since longer cycles are more likely to arise from accumulated noise and combinatorial effects due to graph size, we focus on small cycles, whose presence provides more direct evidence of instability in evaluation outcomes. Accordingly, our methods are based on truncation on 3-cycle counts and 4-cycle counts.

For any $1 \leq i \leq t$ and $0 \leq j \leq m$, let $C_{i,j}^3$ and $C_{i,j}^4$ denote the numbers of 3-cycles and 4-cycles in $G_{i,j}$, respectively. We then truncate the collection of comparison graphs $\{G_{i,j}\}$ based on these cycle counts, retaining only graphs with relatively few cycles. Specifically, fix $1 \leq K \leq t(m+1)$ and $\lambda = (\lambda_3, \lambda_4) \in \mathbb{R}^2$, and assign each graph $G_{i,j}$ the score $\lambda_3 C_{i,j}^3 + \lambda_4 C_{i,j}^4$. We sort all comparison graphs in increasing order of this score, and retain the first $K$ graphs. We denote the selected graphs by $G_1^\lambda, \ldots, G_K^\lambda$ and the corresponding prompts by $x_1^\lambda, \cdots, x_K^\lambda$.

We then use these selected graphs for the subsequent ranking step. Since they contain fewer short cycles, they are expected to exhibit less local inconsistency and hence provide more stable evidence for ranking.

Ties are handled directly in this cycle-counting step. A tie between two models can be represented by two reciprocal directed edges. This allows us to enumerate directed triangles and quadrilaterals using a unified directed-graph representation. However, cycles formed entirely by mutual ties should not be treated as harmful preference cycles. We therefore also count the pure-tie cycles, denoted by $C_{i,j}^{3,\text{tie}}$ and $C_{i,j}^{4,\text{tie}}$, using only reciprocal edges, and define the bad-cycle counts as

$$C_{i,j}^{3,\text{bad}} = C_{i,j}^3 - C_{i,j}^{3,\text{tie}}, \qquad C_{i,j}^{4,\text{bad}} = C_{i,j}^4 - C_{i,j}^{4,\text{tie}}.$$

The truncation score is then computed from these bad-cycle counts, so that cycles caused only by mutual ties are not penalized as preference inconsistency.

**Remark 1** (Connection to rejection sampling). *Our prompt perturbation and cycle truncation procedure may be viewed as an accept–reject mechanism, conceptually related to rejection sampling. Let $S_K$ denote the $K$-th order statistic of $\{\lambda_3 C_{i,j}^3 + \lambda_4 C_{i,j}^4 : 1 \le i \le t,\ 0 \le j \le m\}$. Then the truncation step can be written as*

$$\delta_{i,j} = \mathbf{1}\{\lambda_3 C_{i,j}^3 + \lambda_4 C_{i,j}^4 \le S_K\},$$

*so that the final ranking is constructed from the retained sample $\{G_{i,j} : \delta_{i,j} = 1\}$. This provides an accept–reject perspective on our method: prompt perturbation generates multiple graph-valued proposals, while cycle truncation retains those that are structurally more consistent and hence potentially closer to the latent preference structure.*

## 3.2 Ranking

In this section, we introduce the details of the ranking procedure. Given $G_1^\lambda, \ldots, G_K^\lambda$, we aggregate the pairwise comparison outcomes over the selected prompts and derive the matrices $\mathbf{W}$ and $\mathbf{T}$. Specifically, for any $1 \le i \ne j \le n$, let

$$\mathbf{W}_{ij} = \left|\{1 \le k \le K : \text{ model } i \succ \text{ model } j \text{ under prompt } x_k^\lambda\}\right| \tag{4}$$

$$\mathbf{T}_{ij} = \left|\{1 \le k \le K : \text{ model } i \sim \text{ model } j \text{ under prompt } x_k^\lambda\}\right| \tag{5}$$

By construction, we have $\mathbf{W}_{ij} + \mathbf{W}_{ji} + \mathbf{T}_{ij} = K$ for all $1 \le i \ne j \le n$. The matrix $\mathbf{W}$ records the pairwise wins, while $\mathbf{T}$ records the ties, and together they summarize the comparison evidence extracted from the retained graphs.

**Bradley–Terry model.** Since the Bradley–Terry model in (2) does not explicitly accommodate ties, we use the modified matrix $\tilde{\mathbf{W}}$ defined by $\tilde{\mathbf{W}}_{ij} = \mathbf{W}_{ij} + \frac{1}{2}\mathbf{T}_{ij}$ for all $1 \le i \ne j \le n$, where each tie is split equally between the two competing models. We then maximize the corresponding log-likelihood

$$\ell(\theta) \propto \sum_{1 \le i < j \le n} \left[-\tilde{\mathbf{W}}_{ij} \log(1 + e^{\theta_j - \theta_i}) - \tilde{\mathbf{W}}_{ji} \log(1 + e^{\theta_i - \theta_j})\right]$$

under the identifiability constraint $\theta_1 = 0$, since shifting all components of $\theta$ by the same constant does not change the optimal solution. The resulting estimator $\hat{\theta} = (\hat{\theta}_1, \ldots, \hat{\theta}_n)$ provides the latent skill scores of the models, and we rank the models according to these estimated scores.

**Davidson model.**    Recall that, in Section 2.2, we introduced the Davidson model, which explicitly incorporates ties. As a result, we can directly maximize the corresponding log-likelihood

$$\ell(\theta) \propto \sum_{1 \leq i < j \leq n} \left[ - \mathbf{W}_{ij} \log \Big( \frac{e^{\theta_i}}{e^{\theta_i} + e^{\theta_j} + \nu \sqrt{e^{\theta_i} e^{\theta_j}}} \Big) - \mathbf{W}_{ji} \log \Big( \frac{e^{\theta_j}}{e^{\theta_i} + e^{\theta_j} + \nu \sqrt{e^{\theta_i} e^{\theta_j}}} \Big) \right.$$
$$\left. - \mathbf{T}_{ij} \log \Big( \frac{\nu \sqrt{e^{\theta_i} e^{\theta_j}}}{e^{\theta_i} + e^{\theta_j} + \nu \sqrt{e^{\theta_i} e^{\theta_j}}} \Big) \right]$$

under the constraint $\nu \geq 0$. The resulting estimator then yields the latent scores $\hat{\theta} = (\hat{\theta}_1, \cdots, \hat{\theta}_n)$ of the models, and we rank the models accordingly.

# 4    Experiments and Analysis

## 4.1    Experimental Setup

Unless otherwise stated, all main experimental results reported in this section are obtained with the half-tie Bradley–Terry ranker described in Section 3. We implement the Davidson model as an alternative tie-aware ranker, but use it only as a supplementary robustness check; its optimization details are given in Appendix A.2.

We evaluate model responses on MT-Bench (Zheng et al., 2023), a widely used multi-turn benchmark for open-ended conversational evaluation. MT-Bench contains 80 manually curated questions, with 10 questions in each of eight categories: writing, roleplay, extraction, reasoning, math, coding, STEM, and humanities/social science. It is designed to assess both multi-turn conversational ability and instruction following (Zheng et al., 2023), and has become a standard testbed in the literature for evaluating chat-oriented LLMs (Bai et al., 2024; Chiang et al., 2024; Laban et al., 2026; Zhang et al., 2025). We conduct our experiments on a set of $n = 20$ target LLMs; see Table 5 in Appendix A.1 for the full list. For each question, we keep the original prompt and generate four additional semantically equivalent perturbations using GPT-5.2 (OpenAI, 2025), which gives five prompt instances per question (see the prompt template in Table 1). The generated perturbations are further filtered by a semantic-equivalence checker, whose prompt and filtering details are given in Appendix C.1. For each prompt instance, we collect pairwise judgments for all $\binom{n}{2} = 190$ model pairs and construct a directed comparison graph. Since MT-Bench contains 8 categories, 10 questions per category, and 5 comparison graphs per question, each judge yields $8 \times 10 \times 5 = 400$ comparison graphs.

Table 1: Prompt template for semantic question perturbation.

| Objective | Template |
|---|---|
| Semantic question perturbation | For the question [QUERIED QUESTION], provide $m$ semantically equivalent paraphrases. |

Following the LLM-as-a-judge paradigm, we use two judge models in our experiments: GPT-5.2 (OpenAI, 2025) (abbreviated as GPT-5 below) and a locally deployed Prometheus-family evaluator, M-Prometheus-14B (abbreviated as Prometheus below). GPT-5 is a strong proprietary frontier model with improved instruction following and performance on complex, open-ended tasks, while M-Prometheus-14B serves as an open judge model in the same evaluation pipeline.

GPT-5 is queried through an API with a single pairwise-comparison prompt. Prometheus is run locally in a position-debiased hybrid mode. For each answer pair, we first run the pairwise prompt twice, once under the original answer order and once under the swapped order. If the two verdicts agree after mapping the reversed-order output back to the original answer identities, we keep that pairwise decision. Otherwise, we fall back to direct scoring under the same rubric and use the higher score as the final verdict, with equal scores recorded as ties. Appendix C.1 provides the exact prompts and decoding settings.

For a graph $G$, truncation is based on the bad-cycle statistics introduced above:

$$S_\mu(G) = C_3^{\text{bad}}(G) + \mu C_4^{\text{bad}}(G).$$

We set $\mu = 1$ by default. The keep-$K$ and $\mu$-sensitivity analyses below vary the retained-graph budget in the usual way. For the main baseline comparison in Table 2, however, we report resampled summaries rather than single point estimates. For our method, we first retain the 25 graphs with the smallest values of $S_\mu(G)$ within each category and then repeatedly draw 20 of these graphs without replacement. The sampling-only control uses the same $25 \rightarrow 20$ protocol on the no-perturb pool. The no-truncation baseline applies ordinary bootstrap to the full graph pool, the score-of-win baseline ranks 40-graph subsamples, and the random baseline averages 10 random 20-graph subsets from the 50-graph category pool. Unless otherwise stated, all reported means and confidence intervals are based on 100 outer resamples. After each resample, the selected graphs are aggregated into the matrices $\mathbf{W}$ and $\mathbf{T}$ defined in (4) and (5), respectively.

To evaluate ranking quality, we compare each task-wise ranking with an external reference ranking derived from a fixed snapshot of the public Arena leaderboard data (Chiang et al., 2024). Our primary metric is the normalized Spearman $\rho$ distance,

$$d_\rho(r, \tilde{r}) = \frac{1 - \rho_S(r, \tilde{r})}{2}, \quad \rho_S(r, \tilde{r}) = 1 - \frac{6 \sum_{i=1}^n (r_i - \tilde{r}_i)^2}{n(n^2 - 1)},$$

which maps identical rankings to 0 and reversed rankings to 1 (Spearman, 1904). Appendix C.2 also reports Kendall tau, Spearman footrule, and Chebyshev distances. We treat these distances as ranking losses in the analysis. We compare the proposed method with several baselines and ablations. The baselines include the single-prompt baseline (reported as *Single→20* in Table 2), the no-truncation baseline (*NoTrunc boot*), the score-of-win baseline (*ScoreWin40*), random subset averaging (*Random50→20×10*), and the sampling-only baseline (*Sample25→20*). We also report two block-wise variants, *BlockTop2* and *BlockTop1*. Here, a block refers to the set of comparison graphs generated from the same original prompt, including the original prompt graph and its perturbed variants. BlockTop2 and BlockTop1 retain at most two or one low-cycle graphs from each such block, respectively, and are included to test whether global top-$K$

selection over-concentrates on a small number of original prompts. Appendix A.3 gives the exact source-pool and resampling rule for each baseline.

## 4.2  Ranking Performance

**Relation between cycle counts and ranking accuracy.**  We first examine whether cycle counts are empirically related to ranking accuracy before applying truncation. For each judge, prompt perturbation on MT-Bench yields 400 untruncated comparison graphs. We sort these graphs by the bad-cycle score $S_1(G) = C_3^{\mathrm{bad}}(G) + C_4^{\mathrm{bad}}(G)$ and partition them into 8 consecutive groups of 50 graphs. For each graph, we fit a Bradley–Terry model to its pairwise outcomes and compute the normalized Spearman distance between the resulting ranking and the Arena-derived reference ranking. Figure 2 plots the mean distance within each group against the corresponding mean value of $\log(1 + S_1(G))$, with 95% bootstrap confidence intervals. Graphs with larger bad-cycle scores tend to have larger ranking distances under both judges, indicating that short-cycle inconsistency is positively associated with ranking error.

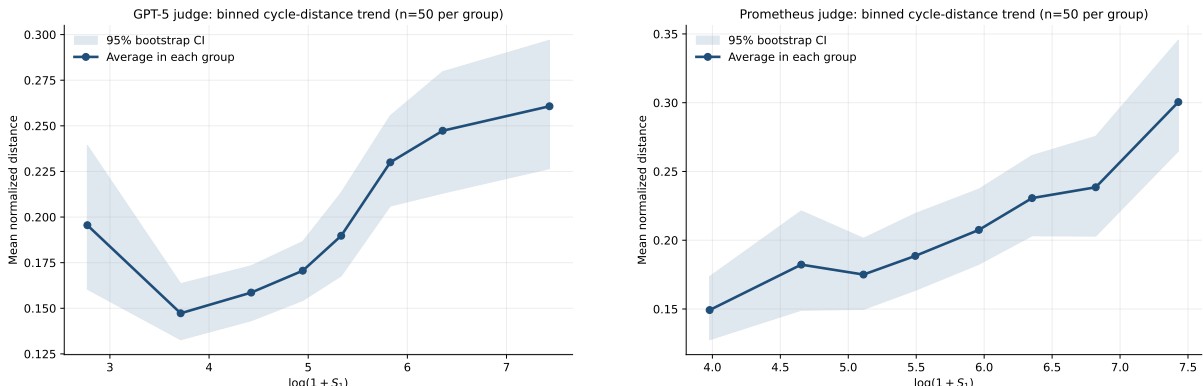

Figure 2: Cycle-based score versus normalized Spearman ranking distance before truncation. The binned trends show a positive association between bad-cycle counts and ranking error for both GPT-5 and Prometheus judges.

**Our method vs. baselines.**  Table 2 reports the mean normalized Spearman $\rho$ distance together with its 95% confidence interval for each judge–task–method cell. This task-wise reporting reflects our task-dependent evaluation setting: each task category is treated as a separate evaluation problem, and the macro-average summarizes performance across categories. Lower values indicate better agreement with the Arena-derived reference ranking. Boldface marks the lowest value within each task column and the lowest macro-average within each judge panel.

Table 2 provides the main baseline comparison in a form that is easier to read quantitatively. Trunc25→20 has the lowest macro-average under both judges: 0.056 for GPT-5 and 0.070 for Prometheus. It also achieves the lowest task-level mean on five of eight tasks under GPT-5 (Coding, Extraction, Math, Roleplay, and Writing) and on four of eight tasks under Prometheus (Coding, Extraction, Math, and Roleplay).

The block-wise variants further examine whether global truncation over-selects perturbations from a small number of original prompts. BlockTop2 and BlockTop1 retain at most two or one graphs from each original-prompt block, respectively. They remain competitive in several tasks,

Table 2: Mean normalized Spearman $\rho$ distance with 95% confidence intervals across 100 resamples. All values are reported in units of $10^{-3}$; for example, 70 means 0.070. Lower values indicate better agreement with the Arena-derived reference ranking.

| | Normalized Spearman $\rho$ distance ($\times 10^{-3}$) | | | | | | | | |
|---|---|---|---|---|---|---|---|---|---|
| Method | Coding | Extraction | Humanities | Math | Reasoning | Roleplay | STEM | Writing | Avg. |
| **GPT-5** | | | | | | | | | |
| Trunc25→20 | 70 ± 1 | **49 ± 2** | 61 ± 1 | **24 ± 1** | 54 ± 1 | 44 ± 1 | 47 ± 1 | **102 ± 1** | **56** |
| BlockTop2 | 73 ± 0 | 105 ± 0 | 53 ± 0 | 32 ± 0 | **37 ± 0** | 65 ± 0 | 51 ± 0 | 142 ± 0 | 70 |
| BlockTop1 | **65 ± 0** | 114 ± 0 | 59 ± 0 | 29 ± 0 | 44 ± 0 | **43 ± 0** | 83 ± 0 | 139 ± 0 | 72 |
| Sample25→20 | 121 ± 1 | 128 ± 2 | 121 ± 1 | 122 ± 1 | 132 ± 2 | 80 ± 1 | 95 ± 1 | 168 ± 1 | 121 |
| NoTrunc boot | 75 ± 2 | 99 ± 4 | 44 ± 2 | 28 ± 1 | 51 ± 2 | 57 ± 1 | **46 ± 1** | 142 ± 3 | 68 |
| ScoreWin40 | 79 ± 2 | 75 ± 2 | **36 ± 2** | 25 ± 1 | 50 ± 1 | 59 ± 2 | 52 ± 1 | 141 ± 2 | 65 |
| Random50→20×10 | 76 ± 1 | 104 ± 2 | 46 ± 1 | 30 ± 1 | 54 ± 1 | 59 ± 1 | 48 ± 1 | 145 ± 1 | 70 |
| Single→20 | 83 ± 0 | 74 ± 0 | 54 ± 0 | 39 ± 0 | 55 ± 0 | 57 ± 0 | 54 ± 0 | 168 ± 0 | 73 |
| **Prometheus** | | | | | | | | | |
| Trunc25→20 | **52 ± 1** | **55 ± 4** | 90 ± 1 | **20 ± 1** | 56 ± 1 | **53 ± 2** | 72 ± 2 | 159 ± 2 | **70** |
| BlockTop2 | 53 ± 0 | 94 ± 0 | 86 ± 0 | **20 ± 0** | 54 ± 0 | 92 ± 0 | **53 ± 0** | 156 ± 0 | 76 |
| BlockTop1 | 54 ± 0 | 137 ± 0 | 91 ± 0 | 30 ± 0 | 70 ± 0 | 93 ± 0 | 56 ± 0 | 159 ± 0 | 86 |
| Sample25→20 | 78 ± 1 | 182 ± 3 | 126 ± 1 | 99 ± 1 | 111 ± 1 | 99 ± 1 | 102 ± 1 | 174 ± 2 | 122 |
| NoTrunc boot | 66 ± 2 | 118 ± 10 | 88 ± 2 | 41 ± 2 | 57 ± 2 | 79 ± 3 | 59 ± 2 | 159 ± 3 | 83 |
| ScoreWin40 | 60 ± 1 | 84 ± 6 | **85 ± 1** | 35 ± 1 | **51 ± 2** | 69 ± 2 | 55 ± 1 | 152 ± 2 | 74 |
| Random50→20×10 | 69 ± 1 | 132 ± 3 | 90 ± 1 | 43 ± 1 | 62 ± 1 | 80 ± 1 | 60 ± 1 | 162 ± 1 | 87 |
| Single→20 | 83 ± 0 | 111 ± 0 | 86 ± 0 | 49 ± 0 | 98 ± 0 | 87 ± 0 | 113 ± 0 | **132 ± 0** | 95 |

but their macro-average distances are generally higher than Trunc25→20, suggesting a trade-off between prompt-level diversity and retaining the strongest low-cycle comparison graphs.

The comparison between Trunc25→20 and Sample25→20 shows the value of semantic perturbation with cycle-aware filtering. Sample25→20 is worse on every task for both judges, and its macro-average is more than twice as large as that of Trunc25→20 under GPT-5 (0.120 vs. 0.056) and clearly larger under Prometheus as well (0.121 vs. 0.070). This pattern suggests that simply repeating stochastic sampling is not enough; the quality of the retained graphs matters.

The comparison with NoTrunc boot isolates the contribution of cycle-aware filtering. Both methods use the perturbed-prompt pool, but only Trunc25→20 removes graphs with high short-cycle counts before ranking. Trunc25→20 is better than NoTrunc boot on five of eight tasks for each judge, with especially large gains on Extraction, Roleplay, and Writing for GPT-5, and on Coding, Extraction, Math, and Roleplay for Prometheus. This supports the view that filtering improves the reliability of the retained comparison evidence, rather than merely changing the sample size.

This task-level variation is consistent with domain-specific model strengths: pairwise preferences can reverse across domains, so a single ordering need not hold uniformly across all tasks. The advantage is not uniform across all tasks, so the result should be stated with the right level of caution. Humanities is better served by ScoreWin40 under both judges, Reasoning is slightly better for ScoreWin40 under GPT-5 and Prometheus, STEM favors NoTrunc boot for GPT-5 and ScoreWin40 for Prometheus, and Writing under Prometheus is lowest for Single→20. In other words, Trunc25→20 is the most consistent method overall, but not the best method on every single category.

Finally, ScoreWin40 remains a competitive baseline. Its macro-average is the second-best under both judges, and it wins several individual tasks. This suggests that strong aggregation can already recover useful ranking information, but the best overall results still come from combining semantic perturbation with cycle-aware filtering before the final ranking step.

We report additional experiments in Appendix C.2. In particular, we evaluate the method on HumanEval and MATH, and also consider a judge-prompt perturbation setting where comparison graphs are generated by varying the judge instruction rather than the evaluated user prompt. These experiments provide complementary evidence beyond the main MT-Bench prompt-perturbation setting.

**Different sources of improvement.** To separate the effect of cycle-aware filtering from the effect of using a larger evaluation budget, we compare truncated and untruncated aggregation under matched source-graph budgets. For a budget of $B$ source graphs per task, both methods use the same number of pairwise judge calls, namely $B\binom{n}{2}$. The only difference is whether the retained graphs are selected by the cycle score $S_\mu(G)$ or used without truncation.

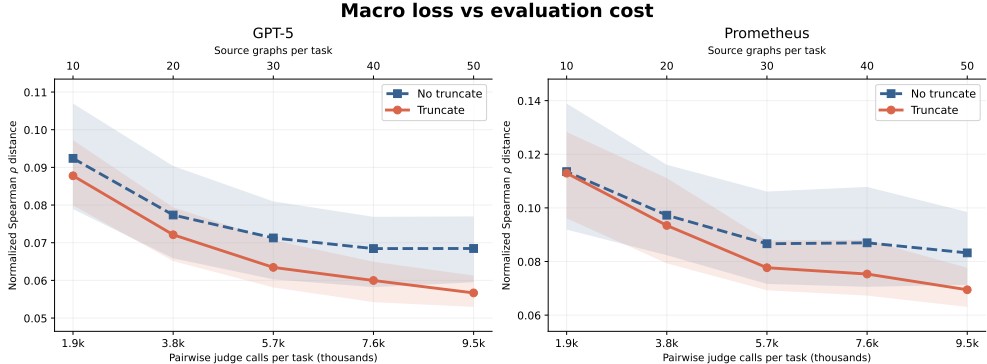

Figure 3: Macro-average normalized Spearman distance versus evaluation cost. Truncated and untruncated aggregation are compared under the same source-graph budget and the same number of pairwise judge calls.

Figure 3 shows that cycle-aware truncation consistently achieves lower macro-average ranking loss than no truncation under the same evaluation cost for both GPT-5 and Prometheus. This indicates that the improvement is not simply due to using more perturbed prompts or more judge calls. Instead, selecting graphs with lower cycle inconsistency contributes additional gain by improving the quality of the retained comparison evidence.

## 4.3 Diagnostics and Sensitivity Analyses

**Model diagnostics.** For each judge–task–selection setting, we fit the half-tie Bradley–Terry model on the selected comparison graphs and evaluate it on held-out comparison graphs. Let $\widetilde{W}_{ij} = W_{ij} + \frac{1}{2}T_{ij}$ denote the half-tie win count. After fitting the Bradley–Terry scores on the selected data, the predicted held-out win probability is $\widehat{p}_{ij} = \frac{\exp(\widehat{\theta}_i)}{\exp(\widehat{\theta}_i)+\exp(\widehat{\theta}_j)}$. The Bradley–Terry held-out loss is the average negative log-likelihood

$$\mathcal{L}_{\mathrm{BT}}^{\mathrm{heldout}} = -\frac{1}{N_{\mathrm{heldout}}} \sum_{i<j} \left[ \widetilde{W}_{ij}^{\mathrm{heldout}} \log \widehat{p}_{ij} + \widetilde{W}_{ji}^{\mathrm{heldout}} \log(1 - \widehat{p}_{ij}) \right].$$

As a nonparametric pairwise baseline, we also compute an empirical held-out loss using the pairwise win rate estimated from the selected data, $\widehat{q}_{ij} = \frac{\widetilde{W}_{ij}^{\mathrm{selected}}}{\widetilde{W}_{ij}^{\mathrm{selected}}+\widetilde{W}_{ji}^{\mathrm{selected}}}$, with a small smoothing

constant used when needed to avoid zero probabilities. Its held-out loss is

$$\mathcal{L}_{\text{emp}}^{\text{heldout}} = -\frac{1}{N_{\text{heldout}}} \sum_{i<j} \left[ \widetilde{W}_{ij}^{\text{heldout}} \log \widehat{q}_{ij} + \widetilde{W}_{ji}^{\text{heldout}} \log(1 - \widehat{q}_{ij}) \right].$$

We report

$$\Delta_{\text{cv}} = \mathcal{L}_{\text{BT}}^{\text{heldout}} - \mathcal{L}_{\text{emp}}^{\text{heldout}}.$$

Thus, a positive value of $\Delta_{\text{cv}}$ means that the Bradley–Terry score model predicts held-out pairwise comparisons worse than the empirical pairwise baseline, which we treat as evidence of possible model misspecification.

For each task, we combine this held-out diagnostic with three in-sample goodness-of-fit diagnostics: normalized deviance, maximum standardized residual, and a cyclic residual (Jiang et al., 2011). We compute the diagnostics using the same aggregated matrices $W, T$ and half-tie Bradley–Terry fit $\widehat{\theta}$ as in Section 3.2. The first signal is the normalized deviance,

$$\frac{2}{N_{\text{obs}} - (n-1)} \sum_{i<j} \left[ \widetilde{W}_{ij} \log \frac{\widetilde{W}_{ij}}{N_{ij} P_{i \succ j}(\widehat{\theta}_i, \widehat{\theta}_j)} + \widetilde{W}_{ji} \log \frac{\widetilde{W}_{ji}}{N_{ij} P_{i \prec j}(\widehat{\theta}_i, \widehat{\theta}_j)} \right],$$

which compares the Bradley–Terry fit with the pairwise saturated binomial model. The second signal is the maximum standardized residual,

$$\max_{i<j} \left| \frac{\widetilde{W}_{ij} - N_{ij} P_{i \succ j}(\widehat{\theta}_i, \widehat{\theta}_j)}{\sqrt{N_{ij} P_{i \succ j}(\widehat{\theta}_i, \widehat{\theta}_j) P_{i \prec j}(\widehat{\theta}_i, \widehat{\theta}_j)}} \right|.$$

The third signal follows the Hodge decomposition view of Jiang et al. (2011, Theorems 2 and 3). Let $Y_{ij} = \frac{W_{ij} - W_{ji}}{N_{ij}}$. We project $Y$ onto score-difference flows by $\widehat{s} \in \arg\min_s \sum_{i<j} N_{ij} \{Y_{ij} - (s_i - s_j)\}^2$, and use the normalized residual energy

$$\frac{\sum_{i<j} N_{ij} \{Y_{ij} - (\widehat{s}_i - \widehat{s}_j)\}^2}{\sum_{i<j} N_{ij} Y_{ij}^2}$$

as the cyclic residual. This is the part of the comparison flow not explained by any global score vector. Each diagnostic produces a warning if it crosses its pre-specified threshold. We label a task as *likely misspecified* if at least two diagnostics raise warnings, as *possible misspecification* if exactly one diagnostic raises a warning, and as *no strong warning* if none of the diagnostics raises a warning. Table 3 reports the number of tasks in each category, together with the average $\Delta_{\text{cv}}$. The diagnostic is applied separately to each judge, selection rule, and MT-Bench category. Given the retained graph set $S$, we aggregate its comparisons, treat ties as half wins, and fit the Bradley–Terry model. We then compute four warning signals: the Bradley–Terry deviance relative to a pairwise saturated binomial model, localized pairwise residuals, the Hodge-style cyclic residual, and a grouped held-out prediction gap $\Delta_{\text{cv}}$.

Table 3 shows that truncation reduces Bradley–Terry misspecification warnings. For GPT-5, likely misspecified tasks drop from 2 under no truncation to 0 after truncation. For both judges, block top-$K$ yields the most tasks with no strong warning. The negative $\Delta_{\text{cv}}$ values further indicate that the Bradley–Terry projection is not worse than the empirical pairwise

Table 3: Bradley–Terry model diagnostic summary. "Likely", "Possible", and "No strong" count the number of tasks in each diagnostic category. A positive $\Delta_{\mathrm{cv}}$ indicates worse held-out prediction than an empirical pairwise baseline.

| Judge | Selection | Likely | Possible | No strong | $\Delta_{\mathrm{cv}}$ |
|---|---|---|---|---|---|
| GPT-5 | block top-$K$ | 0 | 2 | 6 | -0.007 |
| GPT-5 | global top-$K$ | 0 | 3 | 5 | -0.007 |
| GPT-5 | no truncation | 2 | 2 | 4 | -0.001 |
| Prometheus | block top-$K$ | 0 | 2 | 6 | -0.005 |
| Prometheus | global top-$K$ | 0 | 4 | 4 | -0.004 |
| Prometheus | no truncation | 0 | 3 | 5 | -0.003 |

baseline on held-out comparisons. Thus, truncation makes the retained data more compatible with score-based aggregation. When misspecification genuinely persists after truncation, our method should not be interpreted as recovering a true globally transitive preference order. The resulting ranking is instead a score-based projection of the retained comparisons. We therefore recommend reporting the diagnostic flags, and using richer task-dependent or graph-based aggregation models when the residual cyclic component is substantively important.

We emphasize that these diagnostics address internal consistency and score-model compatibility, not systematic judge bias. A judge with stable position, verbosity, or style bias may still produce an internally consistent comparison graph. We discuss this limitation in Appendix C.1; in particular, our Prometheus evaluation uses an $A/B$ order-swapping procedure to reduce position bias.

**Effect of the keep-$K$ budget under truncation.** We next study how the final ranking changes with the keep-$K$ budget. Figure 4 plots the normalized Spearman $\rho$ distance as a function of $K$, where the truncation score is $C_3^{\mathrm{bad}} + C_4^{\mathrm{bad}}$. The curves exhibit a bias-variance trade-off. When $K$ is too small, the final ranking is inferred from too little evidence, so the distance to the reference ranking is large and unstable. As $K$ increases, the ranking loss falls sharply for both judges; the distance drops sharply before flattening from the smallest budgets to the best region. After that initial drop, the curves flatten out and the marginal gain becomes small.

The best $K$ lies in the middle of the admissible range, around $K \approx 25$ for both judges. This behavior is intuitive. If $K$ is too small, we discard too much information and the aggregated comparison matrix is too sparse to support a stable ranking. If $K$ is too large, we begin to re-introduce graphs with higher bad-cycle counts, which are precisely the graphs most likely to inject inconsistent local evidence. In that regime, the improvement saturates and can even slightly reverse. This pattern provides a practical justification for our default choice $K = 25$: it sits near the bottom of the curve for both judges, while avoiding the unnecessary noise introduced by larger retained budgets.

A second observation is that the same qualitative shape appears for both judges, despite their different absolute error levels. GPT-5 remains uniformly stronger in absolute distance, but both judges favor an intermediate truncation budget rather than either extreme. This suggests that the keep-$K$ effect is a property of the comparison-graph construction itself, not an artifact of one particular judge.

Figure 4: Effect of the keep-$K$ budget under cycle-aware truncation. The ranking loss decreases rapidly when $K$ grows from very small values, reaches its minimum in the middle of the range, and then plateaus or slightly rebounds as noisier graphs are added back.

**Data-driven choice of $K$.** The keep-$K$ budget can also be selected by a simple data-driven rule. We use grouped cross-validation over the original prompts, so that all perturbations of the same prompt are placed in the same fold. Let $\{1, \ldots, t\} = I_1 \cup \cdots \cup I_B$. For fold $b$, the training and validation graph sets are

$$\mathcal{G}_{\text{tr}}^{(b)} = \{G_{i,j} : i \notin I_b,\ 0 \le j \le m\}, \qquad \mathcal{G}_{\text{va}}^{(b)} = \{G_{i,j} : i \in I_b,\ 0 \le j \le m\}.$$

For a candidate keep proportion $\alpha \in (0, 1]$, we retain the $K^{(b)}(\alpha) = \left\lceil \alpha |\mathcal{G}_{\text{tr}}^{(b)}| \right\rceil$ training graphs with the smallest truncation scores $S_\mu(G)$, fit the Bradley–Terry model on them, and evaluate it on $\mathcal{G}_{\text{va}}^{(b)}$ using the negative Bradley–Terry log-likelihood in (3), averaged over the held-out comparisons, with ties split as half a win and half a loss. Denote this validation loss by $L_{\text{va}}^{(b)}(\alpha)$. We then compute

$$\overline{L}_{\text{cv}}(\alpha) = \frac{1}{B}\sum_{b=1}^{B} L_{\text{va}}^{(b)}(\alpha), \qquad s_{\text{cv}}(\alpha) = \left\{\frac{1}{B-1}\sum_{b=1}^{B}(L_{\text{va}}^{(b)}(\alpha) - \overline{L}_{\text{cv}}(\alpha))^2\right\}^{1/2},$$

and choose

$$\widehat{\alpha} = \arg\min_{\alpha}\left\{\overline{L}_{\text{cv}}(\alpha) + s_{\text{cv}}(\alpha)\right\}.$$

The selected keep budget is then $K = \lceil \widehat{\alpha}\, t(m+1)\rceil$ for the corresponding category. Table 4 shows that the data-driven choices are generally in the intermediate range rather than at the extremes. For GPT-5, the selected $K$'s mostly lie between 20 and 30, close to the default $K = 25$. For Prometheus, the selected budgets are often smaller, suggesting that a weaker or noisier judge benefits from more aggressive truncation. These results support the qualitative keep-$K$ pattern above and indicate that the default choice is consistent with a simple cross-validation based selection rule.

Table 4: Data-driven keep-budget selection by grouped cross-validation ($B = 5$).

| Judge | Category | $K$ | Judge | Category | $K$ |
|-------|----------|-----|-------|----------|-----|
| GPT-5 | Coding | 28 | Prometheus | Coding | 18 |
| GPT-5 | Extraction | 25 | Prometheus | Extraction | 40 |
| GPT-5 | Humanities | 10 | Prometheus | Humanities | 13 |
| GPT-5 | Math | 20 | Prometheus | Math | 13 |
| GPT-5 | Reasoning | 30 | Prometheus | Reasoning | 18 |
| GPT-5 | Roleplay | 30 | Prometheus | Roleplay | 13 |
| GPT-5 | STEM | 20 | Prometheus | STEM | 28 |
| GPT-5 | Writing | 20 | Prometheus | Writing | 15 |

The grouped cross-validation rule is not the only possible choice of the keep budget. Other data-driven criteria include validation against a small set of human judgments when such labels are available. We use grouped cross-validation because it requires no additional human annotation and keeps all perturbations from the same original prompt in the same fold.

**Sensitivity to the relative weighting of bad 3-cycles and bad 4-cycles.** To understand how much the truncation rule depends on the precise cycle weighting, we fix $K = 25$ and vary $\mu$ in

$$S_\mu(G) = C_3^{\mathrm{bad}}(G) + \mu C_4^{\mathrm{bad}}(G).$$

Figure 5 plots the resulting normalized Spearman $\rho$ distance against $\mu$ on a log scale. The main observation is that the curves are remarkably flat. Across two orders of magnitude of $\mu$, the final ranking changes only slightly for either judge, and once $\mu$ enters the moderate range the differences become almost negligible.

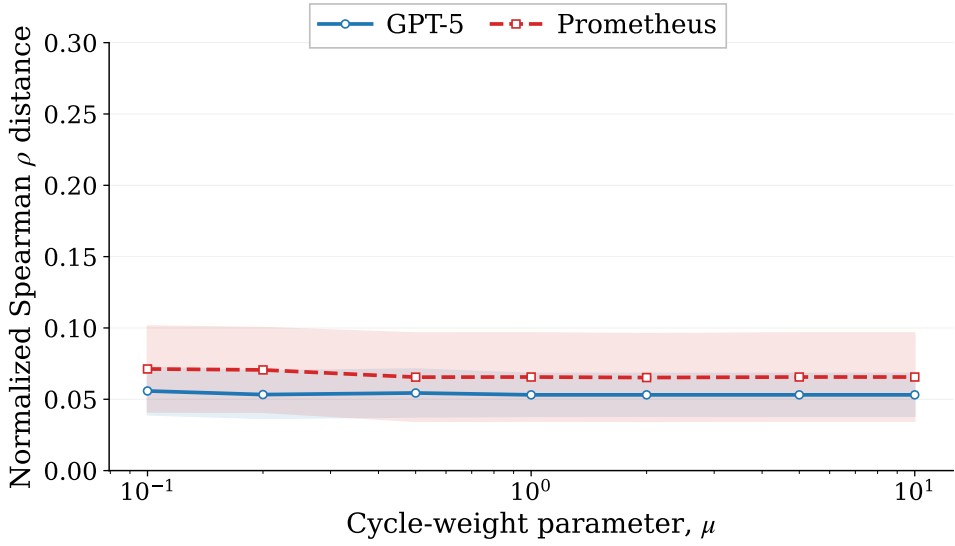

Figure 5: Sensitivity to the relative weighting of bad 3-cycles and bad 4-cycles in the truncation score $S_\mu(G) = C_3^{\mathrm{bad}}(G) + \mu C_4^{\mathrm{bad}}(G)$, with $K = 25$ fixed. The final ranking quality changes only mildly across two orders of magnitude of $\mu$.

This result suggests that the effectiveness of cycle-based truncation does not hinge on a delicate tuning of the relative weights on bad 3-cycles and bad 4-cycles. What matters most is

to remove graphs with substantial short-cycle inconsistency at all; the exact trade-off between triangles and quadrilaterals is secondary. In particular, the near-flat GPT-5 curve and the similarly stable Prometheus curve imply that bad 3-cycles already capture most of the useful signal needed for filtering. Bad 4-cycles can still provide a mild refinement, but they do not materially change the final ranking in this setting.

This observation has an important computational implication. On larger comparison graphs, enumerating 4-cycles is substantially more expensive than counting triangles. Since the ranking distance is almost insensitive to $\mu$, one can often truncate using only $C_3^{\text{bad}}$ or a very coarse approximation to the 4-cycle contribution, and still obtain nearly the same ranking quality. We nevertheless use $\mu = 1$ in the main experiments because it is simple, stable, and performs well across both judges.

# 5    Theoretical Analyses

In this section, we provide a theoretical analysis of a random directed graph model to characterize the interplay among the number of prompt perturbations, the graph size, the success rate of the LLM judge, and the resulting ranking accuracy. Intuitively, increasing the number of prompt perturbations, reducing the graph size, and improving the success rate of the judge model all contribute to a more accurate output ranking. We provide a quantitative analysis for a simple model.

## 5.1    Model Setup

We begin by introducing the probabilistic model. Let $G^*$ denote the latent ground-truth comparison graph, with vertex set $V(G^*) = \{1, 2, \cdots, n\}$ and edge set $E(G^*) = \{(i, j) : 1 \leq i < j \leq n\}$, where $(i, j)$ denotes a directed edge from $i$ to $j$. Equivalently, $G^*$ is the comparison graph induced by the ground-truth ranking $1 \succ 2 \succ \cdots \succ n$. However, due to noise introduced by the prompt perturbation step and possible errors made by the judge model, an observed comparison graph may differ from $G^*$. Specifically, we introduce a parameter $0 < p = p(n) \leq \frac{1}{2}$ to quantify the noise level. For each pair $1 \leq i < j \leq n$, the edge $(i, j)$ in $G^*$ is retained with probability $\frac{1}{2} + p$ and is reversed to $(j, i)$ with probability $\frac{1}{2} - p$. Let $G_1, G_2, \cdots, G_t$ be the observed comparison graphs. Then, for each $1 \leq k \leq t$, we have $V(G_k) = \{1, 2, \cdots, n\}$ and

$$\mathbb{P}\left[(i, j) \in E(G_k)\right] = \frac{1}{2} + p, \quad \mathbb{P}\left[(j, i) \in E(G_k)\right] = \frac{1}{2} - p, \quad \forall 1 \leq i < j \leq n.$$

Moreover, for any $1 \leq k < k' \leq t$, the graphs $G_k$ and $G_{k'}$ are independent conditional on $G^*$.

Note that larger values of $p$ correspond to more reliable judgments, while $p = \frac{1}{2}$ means that the judge model is consistent with the underlying ranking. Our goal is to understand how many comparison graphs are required to recover the latent ranking from observed directed graphs of size $n$ under a given hallucination level characterized by $p$. Specifically, we consider $t$ samples $G_1, G_2, \cdots, G_t$. Let $\tilde{G}$ be the integration graph with $V(\tilde{G}) = \{1, 2, \cdots, n\}$ and

$$(i, j) \in E(\tilde{G}) \text{ if } |\{k : (i, j) \in E(G_k)\}| > \frac{t}{2}, \quad (j, i) \in E(\tilde{G}) \text{ if } |\{k : (i, j) \in E(G_k)\}| \leq \frac{t}{2} \quad (6)$$

for any $1 \leq i < j \leq n$. The graph $\tilde{G}$ provides a natural way to aggregate the information from the $t$ observed directed graphs through pairwise majority voting. We will show in Section 5.2 that if $t$ exceeds a critical threshold, then one can recover the latent ranking from $\tilde{G}$, whereas below another threshold on the other side, the graph $\tilde{G}$ contains a directed cycle, which implies a contradiction with the correct rank.

## 5.2  Theoretical Guarantees

In this subsection, we present theoretical guarantees for the random directed graph model introduced in Section 5.1. Specifically, the following theorem characterizes the sample size required for recovering the latent rank via the majority-vote integration graph $\tilde{G}$. In addition to the positive result showing that the correct rank can be recovered using the majority vote from $\tilde{G}$, we also establish a negative result under which the graph $\tilde{G}$ contains a directed cycle with high probability.

**Theorem 1.** *For any constant $\epsilon > 0$, if $t \geq \frac{(4+\epsilon) \log n}{\log(1/(1-4p^2))}$, then*

$$\mathbb{P}\left[\tilde{G} = G^*\right] = 1 - o(1) \ as \ n \to \infty.$$

*On the other hand, if $t \leq \frac{(4-\epsilon) \log n}{\log(1/(1-4p^2))} \wedge n^{\epsilon/2}$, then the graph $\tilde{G}$ contains at least one directed triangle with probability $1 - o(1)$ as $n \to \infty$. Moreover, under this condition, we have $\mathbb{E}[N_3] = \omega(1)$, where $N_3$ denotes the number of directed triangles in $\tilde{G}$.*

The proof of Theorem 1 is deferred to Appendix B.1. In view of Theorem 1, we derive the sample size required for recovering the latent rank, and also provide a corresponding lower bound under which recovery becomes challenging. We note that the positive and negative results are tight up to constants when $p = n^{-o(1)}$, and tight in rate when $p = n^{-\Theta(1)}$, where smaller values of $p$ correspond to more severe hallucination.

Indeed, our results can be extended to the inhomogeneous setting, where $(i, j)$ in $G^*$ is retained with probability $\frac{1}{2} + p_{ij}$ and reversed to $(j, i)$ with probability $\frac{1}{2} - p_{ij}$ for any $1 \leq i < j \leq n$. This scenario corresponds to the case where the noise level varies across pairs of models. See Remark 3 in Appendix B.1 for details on the theoretical guarantees. Moreover, Theorem 1 is not tied to the total-order assumption. The notation $1 \succ 2 \succ \cdots \succ n$ is used only as a convenient way to describe one possible latent graph. The proof is edgewise: for each unordered pair $\{i, j\}$, the observed edge agrees with the latent edge in $G^*$ with probability $1/2 + p$ and is reversed with probability $1/2 - p$. Majority voting therefore recovers each latent edge with high probability, and a union bound over all $\binom{n}{2}$ pairs yields $\tilde{G} = G^*$ in the stated regime. Thus the same exact-recovery argument applies to an arbitrary latent comparison graph $G^*$, including one that is not induced by a globally transitive ranking.

We next consider the scenario where the structurally consistent observations are accessible, retaining only those with no directed triangle. Specifically, let

$$\mathcal{A} \triangleq \{G : G \text{ contains no directed triangle}\}.$$

We study the conditional law $G \mid \mathcal{A}$. Let $\bar{G}_1, \bar{G}_2, \cdots, \bar{G}_t$ denote samples from this conditioned

distribution. We first characterize the distribution of $\bar{G}_i$. Let $\mathcal{S}_n$ denote the set of permutations from $\{1, 2, \cdots, n\}$ to $\{1, 2, \cdots, n\}$. Indeed, since $\bar{G}_i$ contains no directed triangle, there exists a permutation $\pi_i$ such that $(\pi_i(r), \pi_i(s)) \in E(\bar{G}_i)$ for all $1 \leq r < s \leq n$ (see Lemma 4 in Appendix B.2). Therefore, $\bar{G}_i$ is completely determined by $\pi_i$, and it remains to characterize the distribution of $\pi_i$. This reduction to permutations is closely related to the preference model studied by Liu et al. (2025). In their linear-preference setting, each individual preference is a strict ranking over the $n$ responses, so the sample space consists of the $n!$ possible total orders. In our truncated model, conditioning on the absence of directed triangles has an analogous structural effect: each retained tournament becomes transitive and is therefore also represented by one of the $n!$ total orders. Thus, cycle truncation can be viewed as restricting the comparison-graph support to the same type of total-order support used in scalar preference models. Moreover, by the definition of conditional probability, the probability of observing $\pi_i$ under the conditioned law is exactly the probability of the corresponding graph under the original model, divided by the total probability of all triangle-free graphs. More precisely, let

$$\text{inv}(\pi) = |\{(r, s) : 1 \leq r < s \leq n, \pi(r) > \pi(s)\}|$$

denote the number of inversions of $\pi_i$. Then

$$\mathbb{P}\left[\pi_i \mid \mathcal{A}\right] = \frac{(\frac{1}{2}+p)^{\binom{n}{2}-\text{inv}(\pi_i)}(\frac{1}{2}-p)^{\text{inv}(\pi_i)}}{\sum_{\sigma \in \mathcal{S}_n}(\frac{1}{2}+p)^{\binom{n}{2}-\text{inv}(\sigma)}(\frac{1}{2}-p)^{\text{inv}(\sigma)}} = \frac{q^{\text{inv}(\pi_i)}}{\sum_{\sigma \in \mathcal{S}_n} q^{\text{inv}(\sigma)}}, \quad q = \frac{\frac{1}{2}-p}{\frac{1}{2}+p}.$$

This is precisely the Mallows distribution (Mallows, 1957), and we denote it as $\text{Mallows}_n(q)$. It follows from (Stanley, 2011, Corollary 1.3.13) that $\sum_{\sigma \in \mathcal{S}_n} q^{\text{inv}(\sigma)} = \prod_{r=1}^{n} \frac{1-q^r}{1-q}$. Indeed, for any triangle-free graph $G$ under the conditioned Mallows law, the edge orientation probability between $i$ and $j$ depends only on $j-i$ for any $1 \leq i < j \leq n$; see Proposition 5 in Appendix B.2. In particular, when $j-i = 1$, the probability that $(i, j) \in E(G)$ remains $\frac{1}{2}+p$, while this probability increases with $j - i$. Intuitively, this makes the majority vote $\tilde{G}$ based on $\bar{G}_1, \bar{G}_2, \cdots, \bar{G}_t$ more informative for recovering the ground-truth graph $G^*$. We now state the corresponding upper bound on the number of graphs required for exact recovery of $G^*$.

**Theorem 2.** *Under conditional law $G \mid \mathcal{A}$, for any constant $\epsilon > 0$, if $t \geq \frac{(2+\epsilon)\log n}{\log(1/(1-4p^2))}$, then*

$$\mathbb{P}\left[\tilde{G} = G^*\right] = 1 - o(1) \text{ as } n \to \infty.$$

The proof of Theorem 2 is deferred to Appendix B.2. Comparing Theorems 1 and 2, we see that, after truncating to graphs with no directed triangle, the number of graphs required to recover $G^*$ from $\tilde{G}$ decreases from $\frac{(4+\epsilon)\log n}{\log(1/(1-4p^2))}$ to $\frac{(2+\epsilon)\log n}{\log(1/(1-4p^2))}$. This shows that cycle-based truncation is beneficial even under a simple stylized model: by filtering out comparison graphs with local inconsistencies, one can reduce the sample complexity of exact recovery.

Indeed, in order to obtain $\frac{(2+\epsilon)\log n}{\log(1/(1-4p^2))}$ triangle-free graphs, one may need substantially more than $\frac{(4+\epsilon)\log n}{\log(1/(1-4p^2))}$ original graphs before conditioning on the no-triangle event. From this perspective, direct truncation may appear statistically inefficient, since a potentially large fraction of the original samples is discarded. However, in the regime of LLM evaluation, generating

additional comparison graphs via prompt perturbation is relatively cheap, whereas obtaining high-quality domain prompts is often the real bottleneck. Therefore, despite its low sample efficiency under this stylized model, our truncation-based procedure can still be well suited to practical LLM evaluation. Moreover, one may consider soft truncation rather than imposing the hard constraint of excluding all graphs with directed triangles, since a small number of triangles may not substantially affect the global comparison structure. For example, one may truncate to graphs whose numbers of directed triangles are at most $N$. The two extreme cases recover our previous results: $N = \infty$ corresponds to Theorem 1, while $N = 0$ corresponds to Theorem 2. For intermediate values $N \in (0, \infty)$, one may expect a trade-off between structural consistency and sample efficiency, potentially interpolating between the sample complexities $\frac{(2+\epsilon)\log n}{\log(1/(1-4p^2))}$ and $\frac{(4+\epsilon)\log n}{\log(1/(1-4p^2))}$. A precise characterization of this trade-off is an interesting open problem, which we leave for future work.

It is also important to distinguish the roles of the two theoretical settings. Theorem 1 is an edgewise recovery result and can be interpreted for a general latent comparison graph. By contrast, Theorem 2 conditions on the retained graph having no directed triangle. Under this condition, the support is restricted to transitive tournaments, equivalently the $n!$ possible total orders. Therefore, if the latent preference structure is genuinely non-transitive, the truncated model should not be interpreted as exactly recovering that cyclic structure. Instead, an additional irreducible term may remain, and the resulting ranking could be viewed as an approximation to the latent comparison relation under the chosen loss or likelihood.

We finally discuss the maximum likelihood estimator (MLE) under the two models. Under the untruncated model in Theorem 1, the parameter is the latent graph $G^*$. Conditional on $G^*$, the observed graphs $G_1, \ldots, G_t$ are independent, and for each pair $1 \leq i < j \leq n$, the observed orientation agrees with that in $G^*$ with probability $\frac{1}{2} + p$ and is reversed with probability $\frac{1}{2} - p$. Therefore, the likelihood of $G^*$ is

$$L(G^*) = \prod_{1 \leq i < j \leq n} \left(\tfrac{1}{2} + p\right)^{N_{ij}(G^*)} \left(\tfrac{1}{2} - p\right)^{t - N_{ij}(G^*)},$$

where $N_{ij}(G^*)$ denotes the number of observed graphs whose orientation on the pair $\{i, j\}$ agrees with that in $G^*$. Since the likelihood factorizes over pairs, the MLE is obtained by maximizing each pairwise contribution separately. Equivalently, if

$$N_{ij} \triangleq \left|\{1 \leq k \leq t : (i, j) \in E(G_k)\}\right|, \tag{7}$$

then the MLE chooses $(i, j) \in E(G^*)$ when $N_{ij} > t/2$, and chooses $(j, i) \in E(G^*)$ when $N_{ij} < t/2$, with arbitrary tie-breaking when $N_{ij} = t/2$. Therefore, the integration graph $\tilde{G}$ defined by majority vote is exactly an MLE in this model.

Under the truncated model in Theorem 2, the distribution is characterized by the latent permutation $\pi \sim \text{Mallows}_n(q)$. Recall that $\pi_i$ denote the corresponding permutation induced by $\bar{G}_i$ and $\mathbb{P}\left[\pi_i \mid \pi^*, q\right] = \frac{q^{\text{inv}(\pi_i, \pi^*)}}{\sum_{\sigma \in \mathcal{S}_n} q^{\text{inv}(\sigma)}}$, where $\text{inv}(\pi_i, \pi^*)$ denotes the inversion number between $\pi_i$

and $\pi^*$. Consequently, the likelihood is given by

$$L(\pi^* \mid \pi_1, \cdots, \pi_t) = \prod_{i=1}^{t} \frac{q^{\mathrm{inv}(\pi_i, \pi^*)}}{\sum_{\sigma \in \mathcal{S}_n} q^{\mathrm{inv}(\sigma)}} \propto q^{\sum_{i=1}^{t} \mathrm{inv}(\pi_i, \pi^*)}.$$

Thus the MLE is given by

$$\hat{\pi}_{\mathrm{MLE}} = \operatorname*{argmin}_{\pi \in \mathcal{S}_n} \sum_{i=1}^{t} \mathrm{inv}(\pi_i, \pi).$$

Specifically, we have the following theorem.

**Theorem 3.** *Under conditional law $G \mid \mathcal{A}$, for any constant $\epsilon > 0$, if $t \leq \frac{(2-\epsilon)\log n}{\log(1/(1-4p^2))} \wedge n^{\epsilon/2}$, then*

$$\mathbb{P}\left[\hat{\pi}_{\mathrm{MLE}} \neq \pi^*\right] = 1 - o(1).$$

The proof of Theorem 3 is deferred to Appendix B.3. In view of Theorems 2 and 3, we see that, when $p = n^{-o(1)}$, the estimator $\tilde{G}$ achieves the same exact recovery threshold as the MLE under the truncated model, up to the sharp constant. In this sense, $\tilde{G}$ is statistically competitive with $\hat{\pi}_{\mathrm{MLE}}$. On the other hand, $\tilde{G}$ can be computed by pairwise majority vote in time $O(tn^2)$, whereas a naive exhaustive search for $\hat{\pi}_{\mathrm{MLE}}$ over all $n!$ permutations is computationally prohibitive. This suggests that $\tilde{G}$ provides a simple and computationally efficient alternative while retaining the same recovery threshold in this regime.

## 6    Discussion and Future Directions

In this paper, we propose a new framework for LLM evaluation based on prompt perturbation and cycle truncation. The main idea is to improve the robustness of pairwise evaluation by incorporating perturbations at the prompt level and by controlling local inconsistencies through graph-based cycle truncation. On the theoretical side, we establish guarantees under a random directed graph model, which help explain how the number of perturbations, the graph size, and the reliability of the judge model jointly affect the accuracy of the final ranking. On the empirical side, experiments on real LLM evaluation tasks demonstrate that our method performs favorably and produces more stable and reliable ranking results. We note that lower cycle counts do not universally imply higher evaluation accuracy. Cycle filtering improves graph-level structural consistency, which is an internal reliability criterion rather than a validity guarantee. When the score-based ranking model is misspecified, or when the judge has strong systematic bias, the retained graph may be internally consistent but still misaligned with the desired target. Moreover, if the latent preferences contain stable task-dependent heterogeneity, an irreducible inconsistent component may remain; in this case, the final ranking should be viewed as a score-based projection of the retained comparisons rather than as a true global preference order.

There are several directions for future research.

- *Richer integration methods.* It would be interesting to go beyond the current majority-vote integration rule and develop more refined aggregation procedures that better exploit the structure of pairwise comparison graphs.

- *Broader empirical validation.* It would also be valuable to test the proposed method on a wider range of LLM evaluation tasks, judge models, and benchmark settings to better understand its practical behavior.

- *More realistic theoretical models.* Our current analysis is based on a simple random directed graph model. An important direction is to establish theoretical guarantees under more realistic models that capture heterogeneous prompts, judge bias, and dependence across comparisons.

## Availability

The code is available at https://github.com/soncheinbok/LLMs-Ranking-via-Comparison-Graphs.

## Acknowledgment

The authors thank all reviewers for their valuable comments and suggestions. P. Yang is supported in part by National Key R&D Program of China 2024YFA1015800, Tsinghua University Dushi Program, and High Performance Computing Center, Tsinghua University.

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

# A    Experimental Details

## A.1    Additional Experimental Setup

We list the target LLMs used in our experiments in Table 5, grouped by model family.

We describe the fixed public Arena snapshot used in the experiments and provide the mapping from MT-Bench categories to public Arena categories in Table 6. For categories without an exact counterpart, we use the closest available public category as a proxy.

In the released analysis bundle, the external reference ranking is frozen before resampling and ranker comparisons. All main-table, robustness, and sensitivity runs therefore use the same task-wise reference order rather than re-querying a changing online leaderboard. This is also why the open-code analysis defaults to an embedded reference ranking.

Prompt perturbation is implemented in two stages. The perturbation script first asks the configured API model to over-generate candidate paraphrases in JSON format, with generation temperature 0.7 and a maximum of 12,000 output tokens. Exact duplicates, empty strings, and candidates that normalize to the original prompt are removed. A second deterministic

Table 5: Target LLMs used in the experiments.

| Model family | Models |
| --- | --- |
| Gemma | Gemma 1.1 2B IT; Gemma 2 2B IT; Gemma 2 27B IT |
| GPT | GPT-4.1 nano (2025-04-14 snapshot) |
| Granite | Granite-3.0-2B-Instruct; Granite-3.0-8B-Instruct; Granite-3.1-2B-Instruct |
| Ministral | Mistral-Small-24B-Instruct-2501 |
| Mistral | Mistral-7B-Instruct-v0.1; Mistral-7B-Instruct-v0.2 |
| OLMo | OLMo-7B-Instruct; OLMo-2-0325-32B-Instruct |
| Phi | Phi-3 Mini 4K Instruct; Phi-3 Small 8K Instruct; Phi-3 Medium 4K Instruct; Phi-4 |
| Qwen | Qwen1.5-7B-Chat; Qwen1.5-14B-Chat |
| Others | Zephyr-7B-beta; Starling-LM-7B-beta |

Table 6: Mapping from MT-Bench categories to external reference leaderboards. "Exact" means a direct semantic match; "Proxy" indicates the closest available public LMArena category.

| MT-Bench category | LM Arena category | Type |
| --- | --- | --- |
| Coding | Coding | Exact |
| Extraction | Instruction Following | Proxy |
| Humanities | Writing, Literature & Language | Proxy |
| Math | Math | Exact |
| Reasoning | Hard Prompts (English) | Proxy |
| Roleplay | Multi-Turn | Proxy |
| STEM | Life, Physical & Social Science | Proxy |
| Writing | Creative Writing | Exact |

equivalence-checking call then returns a JSON array of `YES`/`NO` labels, with temperature 0 and a maximum of 600 output tokens. Only candidates marked `YES` are retained. In the reported runs, the configured generator/checker is the GPT-5.2 system mentioned in Section 4.1. The released pipeline therefore separates candidate generation from semantic filtering. The reported MT-Bench analysis uses the fixed five-graph-per-question pool stated in the main text, so each category contains 50 comparison graphs.

## A.2 Implementation Details

When an item is stored as a multi-turn conversation, we evaluate only its first-turn instruction so that all pairwise judgments are based on the same prompt content. Target-model inference is implemented through two backends. For locally deployed models, we use a Hugging Face/Accelerate pipeline with automatic device placement, `bfloat16` precision by default, a limit of 2048 newly generated tokens, and deterministic decoding. For API-based models, we likewise use deterministic decoding, with temperature 0, top-$p = 1$, and a maximum output length of 2048 tokens. In addition, the local pipeline applies a light post-processing step to remove common reasoning tags and stop markers from model outputs before saving them.

Each per-prompt comparison graph is stored as a complete directed graph in which a strict preference $M_i \succ M_j$ is encoded by a single directed edge $i \to j$, while a tie is encoded by the pair of reciprocal edges $i \leftrightarrow j$. For a graph $G$, the implementation exhaustively enumerates all unordered triples and quadruples of vertices and counts every directed orientation that forms

a 3-cycle or 4-cycle. Let $A(G)$ be the adjacency matrix and let $B(G) = A(G) \circ A(G)^\top$ denote the bidirectional-edge matrix. The pure-tie cycle counts $C_3^{\text{tie}}(G)$ and $C_4^{\text{tie}}(G)$ are obtained by applying the same enumeration routine to $B(G)$. The bad-cycle counts are then

$$C_\ell^{\text{bad}}(G) = C_\ell(G) - C_\ell^{\text{tie}}(G), \quad \ell \in \{3, 4\},$$

and the default truncation score is $S_1(G) = C_3^{\text{bad}}(G) + C_4^{\text{bad}}(G)$. For deterministic tie-breaking, graphs are sorted by the pair $(S_1(G), \texttt{source\_order})$. Global truncation keeps the first $K$ graphs in that order. Block-wise truncation first groups graphs by original question id, sorts graphs inside each block by the same score, and then keeps the lowest one or two graphs per block. This is the implementation used for *BlockTop1* and *BlockTop2*.

We report two rankers that differ only in how ties are handled. The first is a half-tie Bradley–Terry ranker. After aggregating the retained graphs into $\mathbf{W}$ and $\mathbf{T}$, we form $\widetilde{\mathbf{W}} = \mathbf{W} + \frac{1}{2}\mathbf{T}$ and minimize the Bradley–Terry negative log-likelihood in an $(n-1)$-dimensional reduced parameterization, fixing the last latent score to zero for identifiability and mean-centering the recovered score vector afterwards. Multiple initializations are tried, and we retain the best converged solution over Newton's method (Turner and Firth, 2012), fixed-point iteration (Yan et al., 2015), minorization–maximization (MM) updates (Hunter, 2004), and quasi-Newton routines (BFGS and L-BFGS-B). For derivative-based runs, the default stopping rule uses $\texttt{maxiter}$=2000 and $\texttt{gtol}$=$10^{-8}$.

The second is an explicit-tie Davidson ranker. It maximizes the Davidson log-likelihood over $n-1$ free log-skill parameters and one free $\log \nu$ parameter, again fixing the last log-skill to zero. Multiple initializations are again tried, and we retain the highest-likelihood converged solution over constrained solvers implemented either in log-parameter space, where $\nu \geq 10^{-12}$ becomes the box constraint $\log \nu \geq \log 10^{-12}$ and is handled by L-BFGS-B or a damped Newton scheme with feasibility safeguarding, or in the original positive parameterization, where fixed-point and MM updates preserve $\lambda_i > 0$ and $\nu > 0$ by construction. The reported Davidson scores are finally rescaled so that the geometric mean of the $\lambda_i$'s equals one. In the ranker-robustness analysis, the same selected graph list is passed unchanged to all five aggregators.

The resampling analysis uses 100 outer resamples with bootstrap seed 20260324. For RANDOM50, each outer resample averages 10 independently drawn 20-graph subsets from the 50-graph category pool. The released local-generation configuration is written for an eight-GPU node and targets $8 \times$ RTX 6000 Ada 48GB for Hugging Face/Accelerate inference. The local Prometheus judge launches one worker per visible free GPU and polls GPU memory/utilization before dispatching category-level jobs. The same seed is also used as the default seed in the keep-$K$, $\mu$-sensitivity, ranker-robustness, and diagnostic scripts unless a script-specific override is supplied.

### A.3   Baseline and Ablation Details

We provide the details for the main baselines and ablations in our experiments in this section. Because Table 2 reports resampled summaries rather than single point estimates, each method is defined by both a source pool and a resampling rule:

- The single-prompt baseline keeps only the original prompt graph for each question. We first choose the best global keep-$K$ on this reduced pool using the same judge-specific ranker as in the main pipeline. Since each category then contains at most ten graphs, the nominal 20-graph subsample usually coincides with the whole selected pool, so the resulting resampling distribution is nearly degenerate.

- The no-truncation baseline skips cycle filtering and applies ordinary bootstrap to the full graph pool in each category. If a category contains $n$ graphs, each resample draws $n$ graphs with replacement and reruns the same likelihood-based ranker.

- The score-of-win baseline draws 40 graphs without replacement, constructs a majority graph from the strict wins within that subset, and ranks models by Copeland-style scores. Exact majorities are encoded as bidirectional edges. Residual score ties are resolved by the same deterministic tie-breaking rule as in the released code.

- The random baseline removes cycle-based selection but keeps the same final subset size as the main method. In each outer resample, we average the ranking loss over 10 independently drawn 20-graph subsets from the 50-graph category pool. This keeps the evaluation budget comparable to the main method while removing the low-cycle selection step.

- The sampling-only baseline replaces semantic perturbation by stochastic replication under the original prompt. We generate five stochastic responses per question (temperature 1.0, top-$p = 1.0$, maximum length 2048), construct the resulting comparison graphs, keep the 25 graphs with the smallest cycle score, and then draw 20 of them without replacement in each resample.

- The block-wise variants operate on prompt families rather than on the full category pool. A block is the set of graphs generated from one original question, including the original graph and its accepted perturbation graphs. *BlockTop2* keeps the two lowest-cycle graphs from each block. Since MT-Bench has 10 questions per category, this yields 20 graphs and can therefore be evaluated directly without an additional 20-out-of-25 resampling step. *BlockTop1* keeps only the lowest-cycle graph from each block. This yields 10 graphs per category. Its reported values are therefore fixed point estimates for the selected set, which is why the corresponding confidence intervals in Table 2 collapse to zero.

## A.4   Details for Additional Analyses

All supplementary analyses reuse the same comparison graphs, the same bad-cycle score $S_\mu(G) = C_3^{\mathrm{bad}}(G) + \mu C_4^{\mathrm{bad}}(G)$, and the same frozen reference ranking as the main pipeline unless stated otherwise. What changes from one analysis to another is only the unit of evaluation and the graph-selection rule. In particular, Figure 2 is a graph-level diagnostic rather than a resampled leaderboard result: for each raw graph, the code fits the same judge-specific report ranker to that graph alone, computes its normalized Spearman distance to the fixed reference ranking, and records the resulting pair $(S_1(G), d_\rho)$. The plotted trend is then drawn from these single-graph records after applying the $\log(1 + \cdot)$ transform to the cycle score and binning on that log scale,

so it should be read as a descriptive summary of how cycle burden and ranking error move together, not as another $25 \rightarrow 20$ estimate.

The cost-matched curve is built from the same perturbation outputs by restricting the available graph ids before any selection is done. Keeping graph ids $1, \ldots, k$ yields $10k$ source graphs per category, so the five reported budgets are $10, 20, 30, 40, 50$; the 10-graph budget is exactly the `graph_id` $= 1$ slice. The truncation arm then applies the same selection rule at a matched ratio: it first keeps about one half of the inspected pool by the cycle score and then samples about 80% of that retained pool without replacement, which gives the sequence $5 \rightarrow 4$, $10 \rightarrow 8$, $15 \rightarrow 12$, $20 \rightarrow 16$, and $25 \rightarrow 20$. The no-truncation arm uses the same inspected pool but bootstraps the whole pool with replacement. Thus, at each budget, the two arms differ only in whether low-cycle graphs are selected before aggregation; the number of inspected graphs, and therefore the number of pairwise judge calls, is the same.

The robustness and diagnostic runs are organized in the same way. Whenever the goal is to compare ranking methods rather than graph-selection rules, the code first fixes one selected graph list for each judge–category–baseline–repeat cell and then feeds that identical list to Bradley–Terry, Davidson, Copeland, Rank Centrality, and HodgeRank. The Bradley–Terry diagnostic script uses the same selected graph families and evaluates them from four angles: saturated-model deviance, standardized pairwise residuals, Hodge residual energy, and grouped held-out prediction. In that script, global top-$K$ means the category-level keep-25 rule, block top-$K$ means keeping the two lowest-cycle graphs from each original-prompt family, and no truncation means using the whole category pool. The grouped cross-validation split is done by original question id, so all perturbations from one prompt stay in the same fold. In the released runs the held-out comparison uses five folds and compares the Bradley–Terry held-out log loss with an empirical pairwise baseline estimated from the training folds with smoothing constant 0.5. The warning thresholds are exactly those used in the code: deviance $p < 0.05$ together with deviance per degree of freedom at least 1.5, maximum standardized residual at least 3 or more than 10% of pairs with $|Z| > 2$, Hodge residual-energy ratio at least 0.20, and a held-out log-loss gap of at least 0.02. The final labels *likely*, *possible*, and *no strong* correspond to two or more, exactly one, and zero warnings. The keep-$K$ and $\mu$ sweeps are direct reruns of the same truncation pipeline under the report ranker chosen for the main experiments; only $K$ or $\mu$ is changed, with the default grids $K = 3, \ldots, 50$ and $\mu \in \{0.1, 0.2, 0.5, 1, 2, 5, 10\}$.

The data-driven choice of $K$ follows the same grouped split by original prompt family, but the search is carried out over keep proportions $\alpha$ rather than over integer budgets directly. For each candidate $\alpha$, the code keeps the lowest-cycle $\lceil \alpha | \mathcal{G}_{\text{tr}}^{(b)} | \rceil$ training graphs in each fold, fits the ranker, and scores held-out pairwise log loss on the validation prompt families. After averaging over folds, it applies a one-standard-error rule in the conservative form used by the script: it finds the minimum mean validation loss, adds one standard error computed at that minimum, and then chooses the smallest $\alpha$ whose mean loss stays below the resulting threshold. The released run adds one further screen that is easy to miss from the main text: for each candidate $\alpha$, the full selected set is also checked by bootstrap rank stability, using 30 bootstrap repeats and threshold $\tau = 0.10$ on the mean pairwise rank distance; candidates that fail this screen are discarded before the final smallest-$\alpha$ choice is made. The reported $K$ is then the category-wise

integer budget obtained from that selected $\alpha$.

The additional HumanEval and MATH experiments reuse the same analysis code, but there are two different perturbation units. For task-prompt perturbation, the benchmark prompt is rewritten by the same generate–check–select pipeline used for MT-Bench. For judge-prompt perturbation, the target-model responses are kept fixed and only the Prometheus judging instruction is perturbed. The released Prometheus script stores ten candidate judge-prompt rewrites, treats five of them as semantically equivalent, and by default runs only those five equivalent variants; the remaining variants are kept only for diagnostics and are not included in the reported semantic-equivalent results. Variant 0 is the original deployed wording, while the semantic-equivalent perturbation pool is formed by the accepted rewritten variants. Each response-prompt id and judge-prompt variant id is encoded into a single `graph_id`, so the downstream selection and ranking code can treat task-prompt perturbation and judge-prompt perturbation through the same graph interface. Overall, these supplementary analyses do not introduce a new reference ranking or a new selection score: they keep the main pipeline fixed and change only the evaluation unit, the budget rule, or the aggregation family.

# B  Proof of Theorems

## B.1  Proof of Theorem 1

Recall the definition of $\tilde{G}$ in (6). Note that $|\{k : (i,j) \in E(G_k)\}| \sim \mathrm{Bin}(t, \frac{1}{2} + p)$, where Bin denote the binomial distribution. For any $1 \le i < j \le n$, we have that

$$\mathbb{P}\left[(i,j) \in E(\tilde{G})\right] = \mathbb{P}\left[\mathrm{Bin}\left(t, \frac{1}{2} + p\right) > \frac{t}{2}\right] = 1 - \mathbb{P}\left[\mathrm{Bin}\left(t, \frac{1}{2} + p\right) \le \frac{t}{2}\right].$$

On the one hand, it follows from Chernoff (1952) that

$$\mathbb{P}\left[\mathrm{Bin}\left(t, \frac{1}{2} + p\right) \le \frac{t}{2}\right] \le \exp\left(-t D_{\mathrm{KL}}\left(\frac{1}{2} \Big\| \frac{1}{2} + p\right)\right) = \exp\left(-\frac{t}{2} \log\left(\frac{1}{1 - 4p^2}\right)\right),$$

where $D_{\mathrm{KL}}(x\|y) \triangleq x \log \frac{x}{y} + (1 - x) \log \frac{1-x}{1-y}$ denotes the Kullback–Leibler (KL) divergence. Let $\mathcal{E}_{ij}$ denote the event that $(i,j) \in E(\tilde{G})$. Consequently, if $t \ge \frac{(4+\epsilon)\log n}{\log(1/(1-4p^2))}$, by a union bound,

$$\mathbb{P}\left[\bigcup_{1 \le i < j \le n} \mathcal{E}_{ij}^c\right] \le \binom{n}{2} \mathbb{P}\left[\mathcal{E}_{ij}^c\right] = \binom{n}{2} \mathbb{P}\left[(i,j) \notin E(\tilde{G})\right]$$

$$\le \exp\left(2\log n - \frac{t}{2} \log\left(\frac{1}{1 - 4p^2}\right)\right) \le \exp\left(-\frac{\epsilon}{2} \log n\right). \tag{8}$$

Then, we have $(i,j) \in E(\tilde{G})$ for all $1 \le i < j \le n$ with probability at least $1 - n^{-\epsilon/2}$. In this case, the correct rank can be recovered by a simple rank-by-wins algorithm, which orders the vertices according to the numbers of outgoing edges in $\tilde{G}$.

On the other hand, it follows from (Ash, 1990, Lemma 4.7.1) that

$$\mathbb{P}\left[\mathrm{Bin}\left(t, \frac{1}{2} + p\right) \le \frac{t}{2}\right] \ge \frac{1}{\sqrt{2t}} \exp\left(-t D_{\mathrm{KL}}\left(\frac{1}{2} \Big\| \frac{1}{2} + p\right)\right) = \frac{1}{\sqrt{2t}} \exp\left(-\frac{t}{2} \log\left(\frac{1}{1 - 4p^2}\right)\right).$$

If $t \leq \frac{(4-\epsilon)\log n}{\log(1/(1-4p^2))} \wedge n^{\epsilon/2}$, for any $1 \leq i < j \leq n$, we have

$$\frac{n^2}{9}\mathbb{P}\left[\mathcal{E}_{ij}^c\right] \geq \frac{n^2}{9}\frac{1}{\sqrt{2t}}\exp\left(-\frac{t}{2}\log\left(\frac{1}{1-4p^2}\right)\right) = \frac{1}{9\sqrt{2}}\exp\left(2\log n - \frac{\log t}{2} - \frac{t}{2}\log\left(\frac{1}{1-4p^2}\right)\right)$$

$$\geq \frac{1}{9\sqrt{2}}\exp\left(2\log n - \frac{\epsilon\log n}{4} - (2-\frac{\epsilon}{2})\log n\right) = \frac{n^{\epsilon/4}}{9\sqrt{2}}. \tag{9}$$

We then obtain

$$\mathbb{P}\left[\bigcup_{\substack{1\leq i\leq n/3 \\ 2n/3\leq j\leq n}}\mathcal{E}_{ij}^c\right] = 1 - \mathbb{P}\left[\bigcap_{\substack{1\leq i\leq n/3 \\ 2n/3\leq j\leq n}}\mathcal{E}_{ij}\right] \overset{(a)}{=} 1 - (1 - \mathbb{P}\left[\mathcal{E}_{ij}^c\right])^{n^2/9} \overset{(b)}{=} 1 - o(1), \tag{10}$$

where (a) follows from $\mathcal{E}_{ij}$ are independent for all $1 \leq i < j \leq n$; (b) is because $\frac{n^2}{9}\mathbb{P}\left[\mathcal{E}_{ij}^c\right] = \omega(1)$. For any given $1 \leq i \leq \frac{n}{3}$ and $\frac{2n}{3} \leq j \leq n$, since $\mathcal{E}_{ik}$ and $\mathcal{E}_{kj}$ are independent, we have

$$\mathbb{P}\left[\mathcal{E}_{ik} \cap \mathcal{E}_{kj}\right] = \mathbb{P}\left[\mathcal{E}_{ik}\right]\mathbb{P}\left[\mathcal{E}_{kj}\right] \geq \frac{1}{2}\cdot\frac{1}{2} = \frac{1}{4}.$$

Consequently,

$$\mathbb{P}\left[\exists\ i < k < j : \mathcal{E}_{ik} \cap \mathcal{E}_{kj}\right] \geq 1 - \left(1 - \frac{1}{4}\right)^{n/3} = 1 - o(1).$$

Combining this with (10), we conclude that with probability there exists a directed cycle ($i \to k \to j \to i$) if $t \leq \frac{(4-\epsilon)\log n}{\log(1/(1-4p^2))} \wedge n^{\epsilon/2}$.

Indeed, the expected number of directed triangles in $\tilde{G}$ already diverges under the condition $t \leq \frac{(4-\epsilon)\log n}{\log(1/(1-4p^2))} \wedge n^{\epsilon/2}$. Let $\tilde{p} = \mathbb{P}\left[\text{Bin}(t, \frac{1}{2}+p) \leq \frac{t}{2}\right]$. For any $1 \leq i < j < k \leq n$, there are two possible directed triangles:

$$i \to k \to j \to i \text{ and } i \to j \to k \to i.$$

These occurs with probabilities of $\tilde{p}^2(1-\tilde{p})$ and $\tilde{p}(1-\tilde{p})^2$, respectively. Let $N_3$ denote the number of directed triangles in $\tilde{G}$. Consequently,

$$\mathbb{E}\left[N_3\right] = \binom{n}{3}\tilde{p}^2(1-\tilde{p}) + \binom{n}{3}\tilde{p}(1-\tilde{p})^2 = \binom{n}{3}\tilde{p}(1-\tilde{p}).$$

It follows from (9) that $\tilde{p} \geq n^{-2-\epsilon/4}/\sqrt{2}$ when $t \leq \frac{(4-\epsilon)\log n}{\log(1/(1-4p^2))} \wedge n^{\epsilon/2}$ and thus $\mathbb{E}\left[N_3\right] = \omega(1)$.

**Remark 2.** *A direct first-moment calculation for the number of directed triangles suggests a larger threshold. Indeed, since $\mathbb{E}\left[N_3\right] = \binom{n}{3}\tilde{p}(1-\tilde{p})$, and $\tilde{p} \to 0$ in the regime of interest, we have $\mathbb{E}\left[N_3\right] \asymp n^3\tilde{p}$. Using the exponential estimate $\tilde{p} \approx \exp\left(-\frac{t}{2}\log\frac{1}{1-4p^2}\right)$, one is naturally led to the heuristic threshold $t \approx \frac{6\log n}{\log(1/(1-4p^2))}$. Therefore, the condition $t \leq \frac{(4-\epsilon)\log n}{\log(1/(1-4p^2))}$ is strictly stronger than what is needed for the first moment of triangles to diverge. The reason is that in the ordered tournament structure, a single reversed edge can typically generate many directed triangles through intermediate vertices, so the actual existence threshold for directed triangles may occur earlier than the first-moment threshold.*

**Remark 3** (Extension to inhomogeneous settings)**.** *Under the inhomogeneous settings, the edge* $(i, j)$ *in* $G^*$ *is retained with probability* $\frac{1}{2} + p_{ij}$ *and reversed to* $(j, i)$ *with probability* $\frac{1}{2} - p_{ij}$ *for any* $1 \leq i < j \leq n$. *Following a similar argument as* (8) *and* (9), *the recovery of the rank by* $\tilde{G}$ *is possible with probability* $1 - o(1)$ *if* $\sum_{1 \leq i < j \leq n} \exp(-\frac{t}{2} \log(\frac{1}{1-4p_{ij}^2})) = o(1)$, *while there exists a directed triangle with probability* $1 - o(1)$ *if* $\sum_{1 \leq i \leq n/3, 2n/3 < j \leq n} \exp(-\frac{1}{2} \log t - \frac{t}{2} \log(\frac{1}{1-4p_{ij}^2})) = \omega(1)$.

## B.2   Proof of Theorem 2

We first show the following lemma, which shows that no directed triangle implies that $G$ is transitive.

**Lemma 4.** *For a directed graph $G$ in which every pair of distinct vertices is connected by exactly one directed edge, the following are equivalent:*

1. *$G$ contains no directed triangle;*

2. *$G$ contains no directed cycle;*

3. *$G$ is transitive, i.e., there exists a permutation $\pi \in S_n$ such that*

$$(\pi(r), \pi(s)) \in E(G) \qquad \text{for all } 1 \leq r < s \leq n.$$

*Proof.* The implication 3 $\implies$ 1 is immediate.

We prove 1 $\implies$ 2. Suppose $G$ contains a directed cycle. Choose one of minimum length

$$v_1 \to v_2 \to \cdots \to v_m \to v_1.$$

Since $G$ has no directed triangle, necessarily $m \geq 4$. Now consider the edge between $v_1$ and $v_3$. If $v_1 \to v_3$, then

$$v_1 \to v_3 \to v_4 \to \cdots \to v_m \to v_1$$

is a shorter directed cycle, contradicting minimality. If $v_3 \to v_1$, then

$$v_1 \to v_2 \to v_3 \to v_1$$

is a directed triangle, again a contradiction. Hence no directed cycle can exist.

Finally, we prove 2 $\implies$ 3. Assume that $G$ contains no directed cycle. We construct the ordering recursively. Choose $\pi(1)$ to be a vertex with maximum out-degree. We claim that $\pi(1)$ dominates every other vertex. Otherwise, if some $u$ satisfies $u \to \pi(1)$, then maximality of the out-degree of $\pi(1)$ implies that there exists a vertex $w$ such that $\pi(1) \to w$ but $u \nrightarrow w$. Since exactly one directed edge is present between each pair of distinct vertices, we must have $w \to u$, and thus

$$u \to \pi(1) \to w \to u$$

is a directed cycle, a contradiction. Therefore, $\pi(1) \to v$ for all $v \neq \pi(1)$. Removing $\pi(1)$ and repeating the same argument on the remaining graph, we obtain a permutation $\pi$ such that $\pi(r) \to \pi(s)$ for all $1 \leq r < s \leq n$. Hence $G$ is transitive.

□

We then show the following proposition on the edge connecting probability condition on $\mathcal{A}$. Recall that $q = \frac{1-2p}{1+2p}$.

**Proposition 5.** *For any $1 \leq i < j \leq n$, let $d \triangleq j - i$. Under the condition law $G \mid \mathcal{A}$, we have*

$$\alpha_d \triangleq \mathbb{P}\left[(i,j) \in E(G) \mid \mathcal{A}\right] = \frac{1 - (d+1)q^d + dq^{d+1}}{(1 - q^d)(1 - q^{d+1})}.$$

*Moreover, we have $\alpha_1 = \frac{1}{1+q} = \frac{1}{2} + p$ and $\alpha_{d+1} > \alpha_d$ for all $d \geq 1$.*

*Proof.* We first claim that, under the conditioned law, the standardized restriction of $\pi$ to any consecutive block again follows a Mallows distribution with the same parameter $q$. For a block $B = \{i, i+1, \ldots, j\}$, let $\pi|_B$ denote the restriction of $\pi$ to the elements in $B$, and let $\mathrm{st}(\pi|_B) \in \mathcal{S}_{d+1}$ denote its standardization, namely, the permutation obtained from $\pi|_B$ by relabeling the smallest element of $B$ as 1, the second smallest as 2, and so on. We claim that $\mathrm{st}(\pi|_B)$ has the Mallows distribution on $\mathcal{S}_{d+1}$ with parameter $q$. Denote $L = \{1, \ldots, i-1\}$ and $R = \{j+1, \ldots, n\}$.

For any permutation $\pi \in \mathcal{S}_n$ satisfying $\mathrm{st}(\pi|_B) = \tau$, let

$$\sigma_L \triangleq \mathrm{st}(\pi|_L) \in \mathcal{S}_{i-1}, \quad \sigma_R \triangleq \mathrm{st}(\pi|_R) \in \mathcal{S}_{n-j}.$$

Moreover, let $w(\pi) = (w_1, \cdots, w_n) \in \{L, B, R\}^n$ denote the interleaving pattern of the three blocks $L, B, R$ in $\pi$, namely, $w_a = L$ if the $a$-th entry of $\pi$ belongs to $L$, and similarly for $B$ and $R$. For any such interleaving pattern $w$, define

$$X^{LB}(w(\pi)) \triangleq \{(a,b) : 1 \leq a < b \leq n, \ w_a = B, \ w_b = L\},$$

$$X^{LR}(w(\pi)) \triangleq \{(a,b) : 1 \leq a < b \leq n, \ w_a = R, \ w_b = L\},$$

$$X^{BR}(w(\pi)) \triangleq \{(a,b) : 1 \leq a < b \leq n, \ w_a = R, \ w_b = B\}.$$

These three sets correspond to the three possible types of cross-block inversions, since every element of $L$ is smaller than every element of $B$, and every element of $B$ is smaller than every element of $R$. Therefore, for every $\pi \in \mathcal{S}_n$ with $\mathrm{st}(\pi|_B) = \tau$, we have

$$\mathrm{inv}(\pi) = \mathrm{inv}(\tau) + \mathrm{inv}(\sigma_L) + \mathrm{inv}(\sigma_R) + |X^{LB}(w(\pi))| + |X^{LR}(w(\pi))| + |X^{BR}(w(\pi))|.$$

Consequently,

$$\sum_{\pi : \mathrm{st}(\pi|_B)=\tau} q^{\mathrm{inv}(\pi)} = q^{\mathrm{inv}(\tau)} \left( \sum_{\sigma_L \in \mathcal{S}_{i-1}} q^{\mathrm{inv}(\sigma_L)} \right) \left( \sum_{\sigma_R \in \mathcal{S}_{n-j}} q^{\mathrm{inv}(\sigma_R)} \right)$$
$$\cdot \left( \sum_w q^{|X^{LB}(w(\pi))|+|X^{LR}(w(\pi))|+|X^{BR}(w(\pi))|} \right),$$

where $w$ ranges over all possible interleavings of the three blocks $L, B, R$. Here the first factor

depends on $\tau$, while the other three factors are independent of $\tau$. Hence

$$\sum_{\pi:\, \mathrm{st}(\pi|_B)=\tau} q^{\mathrm{inv}(\pi)} = C\, q^{\mathrm{inv}(\tau)}$$

for some constant $C$ independent of $\tau$. After normalization over all $\tau \in \mathcal{S}_{d+1}$, it follows that $\mathrm{st}(\pi|_B)$ again has the Mallows distribution on $\mathcal{S}_{d+1}$ with parameter $q$. Consequently,

$$\alpha_d = \mathbb{P}\left[(i,j) \in E(G) \mid \mathcal{A}\right] = \mathbb{P}_{\mathrm{Mallows}_{d+1}(q)}\left[1 \text{ appears before } d+1\right].$$

Recall that the Mallows distribution is given by $\mathbb{P}\left[\pi \mid \mathcal{A}\right] = \frac{q^{\mathrm{inv}(\pi)}}{\sum_{\sigma \in \mathcal{S}_n} q^{\mathrm{inv}(\sigma)}}$. Let $Z_n(q) \triangleq \sum_{\sigma \in \mathcal{S}_n} q^{\mathrm{inv}(\sigma)}$. Then $Z_n(q) = \prod_{r=1}^{n} \frac{1-q^r}{1-q}$. Let $P_{d+1}$ denote the position of $d+1$ from the left in a Mallows distribution on $\mathcal{S}_{d+1}$. If $P_{d+1} = p_{d+1}$, then the largest element $d+1$ creates $d+1-p_{d+1}$ inversions, and the remaining $d$ labels can be arranged arbitrarily. Therefore, the weight with $P_{d+1} = p_{d+1}$ is $q^{d+1-p_{d+1}} Z_d(q)$. Consequently,

$$\mathbb{P}\left[P_{d+1} = p_{d+1}\right] = \frac{q^{d+1-p_{d+1}} Z_d(q)}{Z_{d+1}(q)} = \frac{(1-q)q^{d+1-p_{d+1}}}{1-q^{d+1}}, \quad 1 \le p_{d+1} \le d+1.$$

Let $P_1$ denote the position of 1 from the left in a Mallows permutation on $\mathcal{S}_d$. If $P_1 = p_1$, then the smallest label 1 has exactly $p_1 - 1$ larger labels to its left, so the total weight of such permutations is $q^{p_1-1} Z_{d-1}(q)$. Consequently,

$$\mathbb{P}\left[P_1 = p_1\right] = \frac{q^{p_1-1} Z_{d-1}(q)}{Z_d(q)} = \frac{(1-q)q^{p_1-1}}{1-q^d}, \quad 1 \le p_1 \le d.$$

Then, we have

$$\mathbb{P}\left[1 \text{ appears before } d+1 \mid P_{d+1} = p_{d+1}\right] = \mathbb{P}\left[P_1 \le p_{d+1} - 1 \mid P_{d+1} = p_{d+1}\right]$$
$$= \sum_{p_1=1}^{p_{d+1}-1} \frac{(1-q)q^{p_1-1}}{1-q^d} = \frac{1-q^{p_{d+1}-1}}{1-q^d}.$$

Consequently, we obtain

$$\alpha_d = \mathbb{P}\left[1 \text{ appears before } d+1\right]$$
$$= \sum_{p_{d+1}=1}^{d+1} \mathbb{P}\left[P_{d+1} = p_{d+1}\right] \mathbb{P}\left[1 \text{ appears before } d+1 \mid P_{d+1} = p_{d+1}\right]$$
$$= \sum_{p_{d+1}=1}^{d+1} \frac{(1-q)q^{d+1-p_{d+1}}}{1-q^{d+1}} \cdot \frac{1-q^{p_{d+1}-1}}{1-q^d} = \frac{1-(d+1)q^d + dq^{d+1}}{(1-q^d)(1-q^{d+1})}.$$

For $d=1$, we have $\alpha_1 = \frac{1}{1+q} = \frac{1}{2} + p$. For any $d \ge 1$, we have

$$\alpha_{d+1} - \alpha_d = \frac{q^d(1-q)}{(1-q^d)(1-q^{d+1})(1-q^{d+2})}\left(d(1-q)(1+q^{d+1}) - 2q(1-q^d)\right).$$

Since $q^r + q^{d+1-r} \le 1 + q^{d+1}$ for any $1 \le r \le d$, summing over $1 \le r \le d$ yields $2\sum_{r=1}^{d} q^r \le$

$d(1 + q^{d+1})$. Then $d(1-q)(1 + q^{d+1}) \geq 2q(1 - q^d)$ and thus $\alpha_{d+1} \geq \alpha_d$ for any $d \geq 1$. Since $0 < q < 1$, we have $\alpha_{d+1} > \alpha_d$ for any $d \geq 1$. $\qquad\square$

*Proof of Theorem 2.* Recall the integration graph $\tilde{G}$ defined in 6. For any $1 \leq i < j \leq n$ with $d = j - i$, note that $|\{k : (i,j) \in E(\bar{G}_k)\}| \sim \text{Bin}(t, \alpha_d)$. Consequently, it follows from Chernoff (1952) that

$$\mathbb{P}\left[(i,j) \notin \tilde{G}\right] \leq \mathbb{P}\left[\text{Bin}(t, \alpha_d) \leq t/2\right] \leq \exp\left(-t D_{\text{KL}}(1/2 \| \alpha_d)\right) = e^{-t I_d},$$

where $I_d \triangleq D_{\text{KL}}(1/2 \| \alpha_d) = \frac{1}{2} \log \frac{1}{4 \alpha_d (1 - \alpha_d)}$. We note that there are $n - d$ pairs $(i,j)$ with $j - i = d$. Consequently, we conclude that

$$\mathbb{P}\left[\tilde{G} \neq G^*\right] \leq \sum_{1 \leq i < j \leq n} \mathbb{P}\left[(i,j) \notin E(\tilde{G})\right] \leq \sum_{d=1}^{n-1}(n-d)e^{-t I_d} \leq n e^{-t I_1} + n^2 e^{-t I_2}, \qquad (11)$$

where the last inequality follows from the fact that $\frac{1}{2} < \alpha_1 < \alpha_2 < \alpha_d < 1$ implies $I_d > I_2$ for all $d \geq 3$. Note that

$$(4\alpha_1(1 - \alpha_1))^2 - 4\alpha_2(1 - \alpha_2) = \frac{4q^2(1-q)^2(2q^2 + 3q + 2)}{(1+q)^4(1+q+q^2)^2} \geq 0.$$

Hence, we have $I_2 = \frac{1}{2} \log(\frac{1}{4\alpha_2(1-\alpha_2)}) \geq \log(\frac{1}{4\alpha_1(1-\alpha_1)}) = 2 I_1$. Combining this with (11), we obtain

$$\mathbb{P}\left[\tilde{G} \neq G^*\right] \leq n e^{-t I_1} + n^2 e^{-2t I_1} \leq n^{-\epsilon} + n^{-\epsilon/2} = o(1),$$

where the last inequality follows from $t \geq \frac{(2+\epsilon)\log n}{\log(1/(1-4p^2))} = \frac{(2+\epsilon)\log n}{2 I_1}$. $\qquad\square$

## B.3  Proof of Theorem 3

For any $1 \leq i \leq n - 1$, denote $\pi^{i,i+1} \in \mathcal{S}_n$ as the permutation such that

$$\pi^{i,i+1}(i) = i + 1, \quad \pi^{i,i+1}(i+1) = i, \quad \pi^{i,i+1}(j) = j, \quad \forall j \neq i, i+1.$$

We show that with probability $1 - o(1)$ there exists $i \in \{1, 3, 5, \cdots, \lfloor n/2 \rfloor\}$ such that $\sum_{k=1}^{t} \text{inv}(\pi_k, \pi^{i,i+1}) \leq \sum_{k=1}^{t} \text{inv}(\pi_k, \pi^*)$. Recall $N_{i,j}$ defined in (7). We note that

$$\sum_{k=1}^{t} \text{inv}(\pi_k, \pi^{i,i+1}) - \sum_{k=1}^{t} \text{inv}(\pi_k, \pi^*) = N_{i+1,i} - N_{i,i+1} = t - 2 N_{i,i+1}.$$

Note that $N_{12}, N_{34}, \ldots$ are independent. This is because, for each $\pi_k$, the relative orders on the disjoint pairs $(1,2), (3,4), \ldots$ depend on disjoint consecutive blocks, whose standardized restrictions are independent under the Mallows distribution by the same factorization argument as in Proposition 5. Since $\pi_1, \ldots, \pi_t$ are i.i.d., the corresponding counts $N_{12}, N_{34}, \ldots$ are independent.

By Proposition 5, $N_{i,i+1} \sim \text{Bin}(t, \frac{1}{2} + p)$. It follows from (Ash, 1990, Lemma 4.7.1) that

$$\mathbb{P}\left[\sum_{k=1}^{t} \text{inv}(\pi_k, \pi^{i,i+1}) \leq \sum_{k=1}^{t} \text{inv}(\pi_k, \pi^*)\right] = \mathbb{P}\left[\text{Bin}(t, \frac{1}{2} + p) \leq \frac{t}{2}\right]$$
$$\geq \frac{1}{\sqrt{2t}} \exp\left(-\frac{t}{2}\log\left(\frac{1}{1-4p^2}\right)\right).$$

Then we have

$$\lfloor n/2 \rfloor \mathbb{P}\left[\sum_{k=1}^{t} \text{inv}(\pi_k, \pi^{i,i+1}) \leq \sum_{k=1}^{t} \text{inv}(\pi_k, \pi^*)\right]$$
$$\geq \frac{\lfloor n/2 \rfloor}{\sqrt{2t}} \exp\left(-\frac{t}{2}\log\left(\frac{1}{1-4p^2}\right)\right)$$
$$\geq \frac{1}{3} \exp\left(\log n - \frac{1}{2}\log t - \frac{t}{2}\log\left(\frac{1}{1-4p^2}\right)\right)$$
$$\geq \frac{1}{3} \exp\left(\log n - \frac{\epsilon}{4}\log n - \left(1 - \frac{\epsilon}{2}\right)\log n\right) = \frac{n^{\epsilon/4}}{3}.$$

Therefore,

$$\mathbb{P}\left[\hat{\pi}_{\text{MLE}} = \pi^*\right] \leq \mathbb{P}\left[\bigcap_{i=1}^{\lfloor n/2 \rfloor} \left\{\sum_{k=1}^{t} \text{inv}(\pi_k, \pi^{i,i+1}) > \sum_{k=1}^{t} \text{inv}(\pi_k, \pi^*)\right\}\right]$$
$$\leq \left(1 - \mathbb{P}\left[\text{Bin}(t, \frac{1}{2} + p) > \frac{t}{2}\right]\right)^{\lfloor n/2 \rfloor} = o(1),$$

which implies $\mathbb{P}\left[\hat{\pi}_{\text{MLE}} \neq \pi^*\right] = 1 - o(1)$.

## C  Full Prompts and Additional Results

### C.1  Full Prompts

This appendix records the exact prompt templates used in the evaluation pipeline and clarifies how the final judge decision is produced. Placeholders such as `<instruction>`, `<answer_A>`, and `<ORIGINAL>` denote task-specific content inserted at runtime. We keep the wording close to the deployed implementation, since even small prompt changes can materially affect judge behavior.

**Paraphrase generation and semantic-equivalence filtering.** To construct semantic perturbations, we ask the generator to return exactly $k$ meaning-preserving rewrites of the original instruction in JSON format.

```
Paraphrase generation prompt

You are a strict paraphrase generator.

Generate exactly <k> paraphrases that are semantically equivalent to the ORIGINAL text.

Rules:
- Preserve meaning exactly (same intent, constraints, numbers, entities, and requirements).
- Keep the SAME language as the ORIGINAL.
```

```
- Do NOT answer the question/instruction; only rewrite it.
- Each paraphrase should be meaning-preserving but phrased differently.
- Output MUST be a JSON array of exactly <k> strings. No extra text.

ORIGINAL:
<<ORIGINAL_START>>
<ORIGINAL>
<<ORIGINAL_END>>
```

Raw paraphrase generation can occasionally introduce mild semantic drift. We therefore apply a second-stage filter and retain only candidates marked YES by a strict equivalence checker.

**Semantic equivalence checking prompt**

```
You are a strict semantic equivalence judge.

For each CANDIDATE, decide if it is semantically equivalent to the ORIGINAL.
Equivalent means: same intent + same constraints + no added/removed requirements + same meaning.

Return format:
- Output MUST be a JSON array of strings of length <num_candidates>.
- Each element must be exactly "YES" or "NO".
- No extra text.

ORIGINAL:
<<ORIGINAL_START>>
<ORIGINAL>
<<ORIGINAL_END>>

CANDIDATES:
1) <candidate_1>
2) <candidate_2>
...
<num_candidates>) <candidate_num_candidates>
```

**Judge prompt.** GPT-5 is invoked with a single pairwise-comparison prompt and no additional task-specific system instruction in the released code path. The model is required to produce a short comparison followed by a final verdict in square brackets.

**GPT-5 pairwise judge prompt**

```
Instruction:
Below is a question and a candidate response from a user. Please act as an impartial
judge and evaluate the quality of the responses provided by two AI assistants to the
user question displayed below. You should choose the assistant that follows the user's
instructions and answers the user's question better. Your evaluation should consider
factors such as the helpfulness, relevance, accuracy, depth, creativity, and level of
detail of their responses. Begin your evaluation by comparing the two responses and
provide a short explanation(NO MORE THAN 512 TOKENS). Avoid any position biases and ensure that the
↪   order in
which the responses were presented does not influence your decision. Do not allow
the length of the responses to influence your evaluation. Do not favor certain names
of the assistants. Be as objective as possible. After providing your brief explanation,
you *must* output the final verdict by strictly following this format: "[A]" if
assistant A is better, "[B]" if assistant B is better, and "[C]" for a tie. Provide a
result exclusively in **square brackets** (e.g., Verdict: [A]).
Question:
<instruction>
Answer A:
<answer_A>
Answer B:
<answer_B>
Judgement:
```

Because local deployment is relatively inexpensive, and prior studies have shown that positional bias can influence LLM-as-a-judge decisions (Shi et al., 2025), our Prometheus evaluator (M-Prometheus-14B) adopts a position-debiased hybrid protocol. For each answer pair, we first run the pairwise prompt twice: once in the original order $(A, B)$ and once after swapping the

two answers $(B, A)$. If the two verdicts agree after mapping the reversed-order output back to the original answer identities, we keep that pairwise decision. Otherwise, we fall back to direct scoring, grade the two answers independently on a 1–5 scale, and select the higher-scoring answer; equal scores are recorded as ties. In the implementation, each answer is truncated to at most 12,000 characters before judging, and generation is deterministic by default (temperature 0, top-$p = 1$, maximum 512 new tokens).



**Prometheus pairwise system prompt**

```
You are a fair judge assistant assigned to deliver insightful feedback that compares individual
↪  performances, highlighting how each stands relative to others within the same cohort.
```

**Prometheus pairwise user prompt**

```
###Task Description:
An instruction (might include an Input inside it), two responses to evaluate, and a score rubric
↪  representing a evaluation criteria are given.
1. Write a detailed feedback that assess the quality of two responses strictly based on the given score
↪  rubric, not evaluating in general.
2. After writing a feedback, choose a better response between Response A and Response B. You should
↪  refer to the score rubric.
3. The output format should look as follows:
"Feedback: (write a feedback for criteria)
[RESULT] (A or B)"
4. Please do not generate any other opening, closing, and explanations.
###Instruction:
<instruction>
###Response A:
<answer_A>
###Response B:
<answer_B>
###Score Rubric:
<pairwise_rubric>
###Feedback:
```

Unless a task-specific rubric is already provided in the source data, we use the following default pairwise rubric.

**Default pairwise rubric used by Prometheus**

```
[Instruction Following, Helpfulness, Relevance, Accuracy, Depth, Clarity, Safety]
Choose the better response by comparing how well it follows the user's instruction, how accurate and
↪  relevant it is, how helpful and sufficiently detailed it is, how clear and coherent it is, and
↪  whether it avoids unsafe or misleading content. Do not prefer a response merely because it is
↪  longer.
```

**Prometheus direct-scoring system prompt**

```
You are a fair judge assistant tasked with providing clear, objective feedback based on specific
↪  criteria, ensuring each assessment reflects the absolute standards set for performance.
```

**Prometheus direct-scoring user prompt**

```
###Task Description:
An instruction (might include an Input inside it), a response to evaluate, a reference answer that gets
↪  a score of 5, and a score rubric representing a evaluation criteria are given.
1. Write a detailed feedback that assess the quality of the response strictly based on the given score
↪  rubric, not evaluating in general.
2. After writing a feedback, write a score that is an integer between 1 and 5. You should refer to the
↪  score rubric.
3. The output format should look as follows:
"Feedback: (write a feedback for criteria)
[RESULT] (an integer number between 1 and 5)"
4. Please do not generate any other opening, closing, and explanations.
###The instruction to evaluate:
<instruction>
###Response to evaluate:
<answer>
```

```
###Score Rubrics:
<direct_rubric>
###Feedback:
```

**Default direct-scoring rubric used by Prometheus**

```
[Instruction Following, Helpfulness, Relevance, Accuracy, Depth, Clarity, Safety]
Score 1: The response fails to address the user's request, is largely incorrect, irrelevant,
↪   incoherent, or unsafe.
Score 2: The response addresses the request only partially and contains major omissions, errors, weak
↪   reasoning, poor clarity, or notable safety issues.
Score 3: The response is acceptable and mostly relevant, but it misses useful details, has minor
↪   factual or reasoning issues, or is only moderately clear/helpful.
Score 4: The response follows the instruction well and is clear, relevant, and mostly accurate, with
↪   only small deficiencies in completeness, precision, or clarity.
Score 5: The response fully satisfies the request and is highly accurate, relevant, coherent,
↪   appropriately detailed, and safe.
```

The hybrid evaluator above can be summarized as follows.

1. Run the pairwise prompt on $(A, B)$ and parse the last valid tag of the form `[RESULT] A` or `[RESULT] B`.

2. Swap the order, run the same pairwise prompt on $(B, A)$, and map the reversed verdict back to the original identities.

3. If the two pairwise verdicts agree, output that verdict.

4. Otherwise, score $A$ and $B$ independently using the direct-scoring prompt.

5. Output `A` if score$(A)$ is greater than score$(B)$, output `B` if score$(B)$ is greater than score$(A)$, and output a tie otherwise.

This description is more faithful to the deployed evaluator than a one-shot pairwise prompt alone: the released Prometheus judge is a two-pass, order-checked pairwise evaluator with a direct-scoring fallback.

## C.2  Additional results

**Structural diagnostics of the comparison graphs.**    Figure 6 reports four descriptive statistics of the comparison graphs: the mean numbers of bad 3-cycles and bad 4-cycles, the mean draw rate, and the mean size of the largest strongly connected component (SCC), all broken down by MT-Bench category and judge.

Two patterns stand out. First, the difficulty profile is judge-dependent. Under GPT-5, Math and Extraction exhibit the largest short-cycle burden, whereas under Prometheus the heaviest inconsistency appears in Extraction and Writing. Humanities is the cleanest category for both judges, with visibly fewer short cycles and smaller SCCs. This suggests that graph inconsistency is not determined by the task alone; it also depends on how a given judge reacts to that task.

Second, prompt perturbation often increases bad 3-cycles, bad 4-cycles, draw rate, and, in several categories, the size of the largest SCC. This does not contradict the stronger downstream rankings reported in the main text. A more plausible interpretation is that semantic perturbation exposes close contests that are genuinely ambiguous, thereby revealing more of the uncertainty already present among similarly strong models. By contrast, the no-perturb setting can look

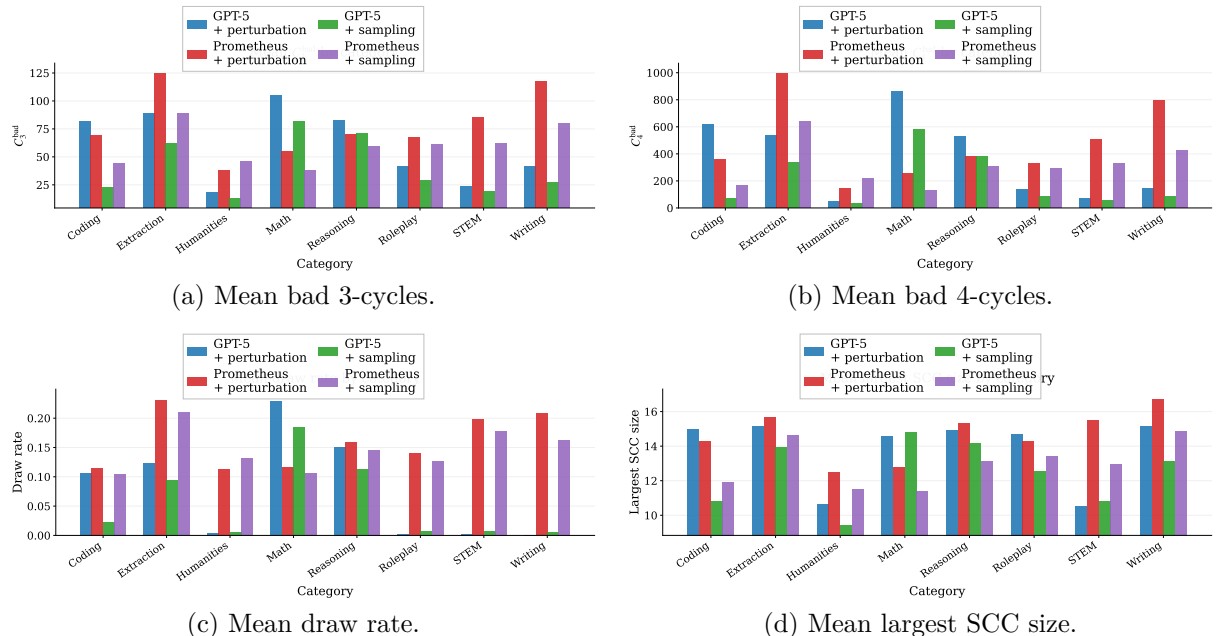

(a) Mean bad 3-cycles.

(b) Mean bad 4-cycles.

(c) Mean draw rate.

(d) Mean largest SCC size.

Figure 6: Descriptive statistics of the comparison graphs by category. Prompt perturbation usually increases short-cycle activity, draw rate, and the size of the largest SCC, indicating that it surfaces more of the latent ambiguity among closely matched models. Despite this increase in raw graph inconsistency, the perturbed-and-truncated pipeline yields better final rankings, so lower bad-cycle counts alone should not be interpreted as better ranking evidence.

artificially clean because stochastic response variation sometimes turns a close comparison into a decisive win or loss. In that regime, the graph contains fewer local contradictions, but it is not necessarily closer to the underlying ranking. The draw-rate panels support this interpretation: in several GPT-5 no-perturb categories, draws nearly disappear, which is more consistent with suppressed ambiguity than with cleaner preference information. At the same time, the largest SCC remains substantial across conditions, so the graphs stay globally connected enough for downstream aggregation to propagate local evidence across models.

**Robustness across ranking distances.** In the main text, we use the normalized Spearman $\rho$ distance as the primary ranking loss. To verify that the conclusions do not depend on this single choice, we also report the normalized Kendall tau distance, the normalized Spearman footrule distance, and the normalized Chebyshev distance. Let $r = (r_1, \ldots, r_n)$ be the estimated ranking and let $r^\star = (r_1^\star, \ldots, r_n^\star)$ be the reference ranking on the same set of $n$ models. We define

$$d_\tau(r, r^\star) = \frac{2}{n(n-1)} \sum_{1 \leq i < j \leq n} \mathbf{1}\Big\{(r_i - r_j)(r_i^\star - r_j^\star) < 0\Big\},$$

$$d_F(r, r^\star) = \frac{\sum_{i=1}^n |r_i - r_i^\star|}{\lfloor n^2/2 \rfloor}, \quad d_\infty(r, r^\star) = \frac{\max_{1 \leq i \leq n} |r_i - r_i^\star|}{n-1}.$$

Lower values indicate rankings closer to the reference order.

Figure 7 shows that the same qualitative pattern holds under all four distances and for both judges. Each curve drops quickly when $K$ grows from very small values, reaches its best region at an intermediate budget, and then flattens or slightly rebounds as additional graphs are retained. The absolute scales differ, as expected. The Chebyshev distance is the noisiest because it depends

only on the single worst-displaced model, whereas Kendall tau and Spearman footrule average disagreement more globally. Even so, the preferred operating region is essentially unchanged: the best performance is consistently achieved at an intermediate keep-$K$ budget close to the value used in the main text. Prometheus remains somewhat higher and noisier in absolute error than GPT-5, but the location and shape of the optimum are similar across judges. This reinforces the conclusion that the benefit of cycle-based truncation is robust to the choice of ranking distance.

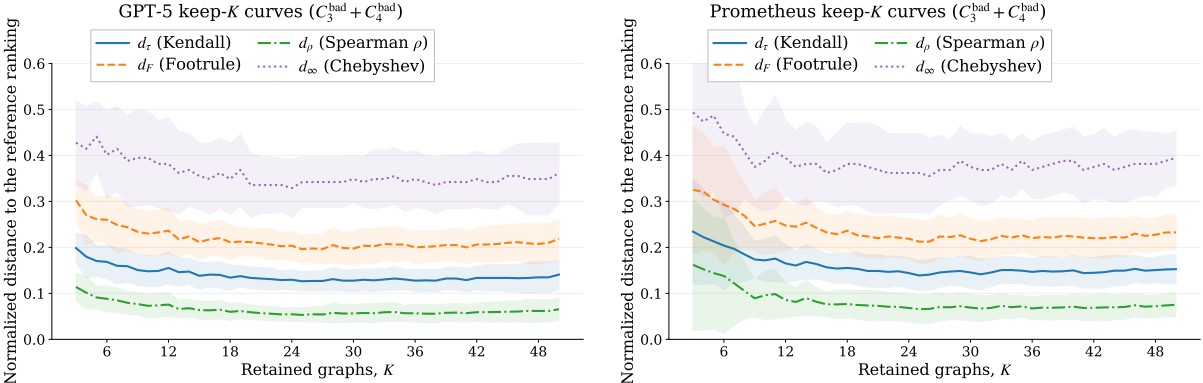

Figure 7: Keep-$K$ curves under four normalized ranking distances. Left: GPT-5 judge. Right: Prometheus judge. The favorable mid-range truncation regime is consistent across Kendall tau, Spearman footrule, Spearman $\rho$, and Chebyshev distances.

**Semantic perturbation versus sampling-only control.** Figure 8 compares semantic prompt perturbation with a sampling-only control under the same cycle-aware truncation score $C_3^{\mathrm{bad}} + C_4^{\mathrm{bad}}$, measured by the normalized Spearman $\rho$ distance. The pattern is the same for both judges. Under GPT-5, the perturbation curve starts around 0.11 and drops to about 0.05–0.07 once $K$ reaches a moderate size, while the sampling-only baseline stays close to 0.11–0.13 throughout. Under Prometheus, the perturbation curve falls from about 0.15 to roughly 0.06–0.08, whereas the sampling-only baseline remains near 0.11–0.13 with little change. The gain from perturbation is therefore not tied to a narrow budget range; it holds across the full keep-$K$ sweep.

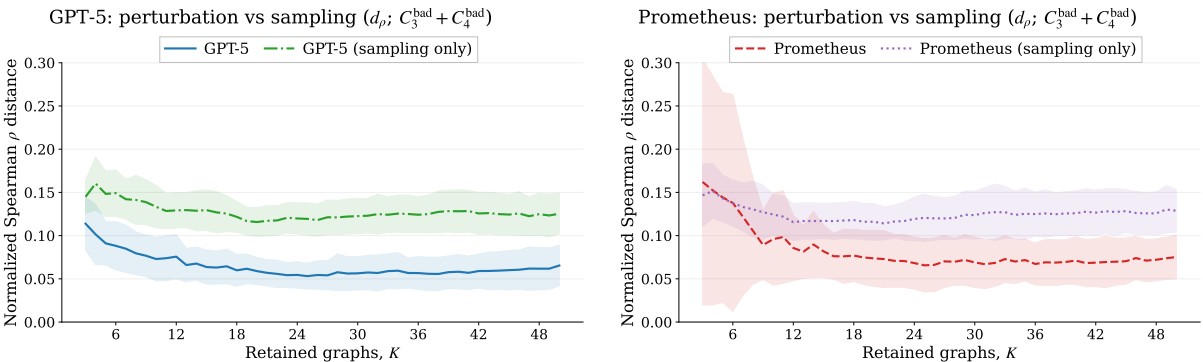

Figure 8: Semantic prompt perturbation versus a sampling-only control under cycle-aware truncation. Lower normalized Spearman distance indicates better agreement with the reference ranking. For both GPT-5 and Prometheus, perturbation stays well below the sampling-only baseline over the full range of retained budgets $K$.

Table 7: Additional HumanEval and MATH experiments. Values are normalized Spearman distances reported in units of $10^{-3}$.

| Method | HumanEval | MATH |
|---|---|---|
| **Task-prompt perturbation** | | |
| Trunc50→40 | **82** ± 1 | **13** ± 1 |
| Block top-2 | 95 ± 0 | 23 ± 0 |
| Block top-1 | 100 ± 0 | 31 ± 0 |
| NoTrunc boot | 92 ± 2 | 21 ± 2 |
| ScoreWin80 | 93 ± 1 | 20 ± 1 |
| Random50→40×10 | 92 ± 1 | 24 ± 1 |
| SingleGraph1 | 96 ± 0 | 18 ± 0 |
| **Judge-prompt perturbation with Prometheus** | | |
| Trunc25→20 | **57** ± 2 | **31** ± 1 |
| Block top-2 | 74 ± 0 | 32 ± 0 |
| Block top-1 | 70 ± 0 | 38 ± 0 |
| NoTrunc boot | 66 ± 2 | 43 ± 2 |
| ScoreWin40 | 65 ± 2 | 37 ± 1 |
| Random50→20×10 | 66 ± 1 | 44 ± 1 |
| SingleGraph1/all10 | 60 ± 0 | 44 ± 0 |

This comparison makes the role of perturbation clear. The improvement does not come from simply generating more graph instances from the same prompt. If repeated sampling were enough, the sampling-only baseline should also improve as more graphs are retained. That is not what the figure shows: for both judges, the sampling-only curves are almost flat. By contrast, the perturbation curves fall quickly at small and moderate $K$ and then level off at a much lower value. The main gain therefore comes from perturbation itself, not from sample count alone.

The shape of the curves also helps explain why perturbation works well with cycle-aware truncation. Most of the improvement appears early, before $K$ becomes large, and the marginal gain becomes small after that. In other words, once the retained set already contains enough informative perturbed graphs, keeping more graphs adds little. The sampling-only control does not show the same pattern, which suggests that its additional graphs contribute little beyond decoding noise.

**Additional HumanEval and MATH experiments.** We further evaluate the proposed cycle-aware filtering pipeline beyond the main MT-Bench setting. For task-prompt perturbation, we apply the same prompt-perturbation and truncation procedure to HumanEval and MATH. We also consider a judge-prompt perturbation setting with Prometheus, where the model responses are fixed and the comparison graphs are generated by perturbing the judge instruction instead of the task prompt. In both cases, the same graph-level selection and ranking pipeline is used. Overall, the proposed truncation method performs best in both settings. For task-prompt perturbation, Trunc50→40 achieves the lowest distance on both HumanEval and MATH. For judge-prompt perturbation, Trunc25→20 also gives the best results on both coding and math. These results provide additional evidence that cycle-aware filtering remains useful beyond the main MT-Bench task-prompt perturbation experiment.

