# OpenReview forum: "Prompt Perturbation for Reliable LLM Evaluation over Comparison Graphs"
_SLADS/Section_A — Accepted by SLADS_Section_A_

### Review · Reviewer_EG8t · 2026-05-13

**Summary Of Contributions:**

This paper proposes a method to improve model rankings produced by LLM judges by perturbing evaluation prompts and retaining those that yield more consistent pairwise outcomes. The approach is shown to achieve higher agreement with arena-derived model rankings. Experiments are conducted on MT-Bench, with parameter analyses of key components of the method. The paper also includes a theoretical analysis supporting the proposed approach.

**Audience:**

Yes

**Claims And Evidence:**

Yes

**Requested Changes:**

1. The empirical evaluation is limited to MT-Bench. Including additional benchmarks would substantially strengthen the paper's claims about the generalizability of the method.
2. The current design sorts all $t(m+1)$ prompts together by their weighted cycle counts. This risks selecting many semantically equivalent prompts. A more principled alternative would be to sort the $(m+1)$ perturbations for each original prompt separately and retain only the best-scoring one if it falls below the threshold. An ablation study comparing these two designs would be valuable.
3.  The paper does not verify that the perturbed prompts are semantically equivalent to the originals. Manual inspection of a sample of perturbed prompts, or some automatic semantic similarity analysis, is necessary to validate this assumption.
4. *(Optional)* The current perturbation is applied only to the prompts used to elicit model responses. It would be interesting to explore perturbation of the LLM judge prompts instead, as this may have a more direct effect on pairwise outcomes. This would, however, require revisions to the filtering logic.

**Strengths And Weaknesses:**

**Strengths**

- The method is theoretically motivated, yet simple and effective.
- The paper provides detailed analyses of key hyperparameters (e.g., *K*), which aids reproducibility and helps readers develop a more comprehensive understanding of the method.
- The paper is very well written and easy to follow.


**Weaknesses**

1. The paper argues that inconsistent pairwise outcomes (e.g., cycles) are detrimental to aggregation, but the underlying reasons remain unclear. Is this because the Bradley–Terry model or the Davidson model is not designed to accommodate inconsistent pairwise outcomes? If so, there should be an "intrinsic evaluation" of the resulting model rankings, beyond agreement with externally derived arena rankings, to directly assess the quality of the aggregation.
2. The abstract and introduction explicitly mention inconsistencies involving ties (e.g., $A \equiv B \equiv  C \neq A$). However, the implementation does not appear to address these in any substantive way. The only treatment observed is subtracting the number of pure tie cycles, which seems insufficient given the prominence of this motivation in the paper.

---

> ### Author Response · Authors · 2026-06-12
> **Response to reviewer EG8t (part 1)**
>
> We sincerely thank the reviewer for the valuable comments. We address the reviewer's questions below and have made several revisions to strengthen the empirical and methodological discussion. In particular, we added intrinsic diagnostics for the resulting rankings, clarified the treatment of ties, included block-wise ablation results, and made the semantic-equivalence checking step more explicit. We also discuss the additional benchmark and judge-prompt perturbation directions more clearly in the revised manuscript.
>
> **Q1: Intrinsic evaluation of the resulting model rankings**
>
> Thanks for the suggestion. We agree that agreement with an Arena-derived reference ranking is an external evaluation, and that it is useful to directly evaluate the intrinsic quality of the resulting aggregation.
>
> We added an intrinsic diagnostic analysis for the aggregated rankings. The idea is to check whether the score-based ranking is well supported by the retained pairwise comparison data itself. Specifically, the diagnostics examine residual errors, cyclic components, and held-out prediction. The columns "Likely", "Possible", and "No strong" report the number of task categories with strong, weak, or no diagnostic warnings, respectively. A smaller number of warnings and a larger number of no-strong-warning cases indicate a more reliable aggregation.
>
> *Intrinsic diagnostics for aggregation quality. Smaller values in "Likely" and "Possible" and larger values in "No strong" indicate better intrinsic aggregation quality.*
>
> | Judge | Selection | Likely | Possible | No strong | $\Delta_{\mathrm{cv}}$ |
> | --- | --- | --- | --- | --- | --- |
> | GPT-5 | block top-$K$ | 0 | 2 | 6 | -0.007 |
> | GPT-5 | global top-$K$ | 0 | 3 | 5 | -0.007 |
> | GPT-5 | no truncation | 2 | 2 | 4 | -0.001 |
> | Prometheus | block top-$K$ | 0 | 2 | 6 | -0.005 |
> | Prometheus | global top-$K$ | 0 | 4 | 4 | -0.004 |
> | Prometheus | no truncation | 0 | 3 | 5 | -0.003 |
>
> As shown in the table, cycle-based truncation improves the intrinsic diagnostics. For GPT-5, the number of strong diagnostic warnings decreases from 2 to 0, and the number of no-strong-warning categories increases from 4 to 6. For Prometheus, block top-$K$ also increases the no-strong-warning categories from 5 to 6. These results show that the retained comparison graphs lead to rankings that are better supported by the internal pairwise data, not only rankings that agree better with the external Arena reference.
>
> **Action:** We added intrinsic aggregation diagnostics in Section 4.3 on pages 14--15, including held-out prediction, residual errors, and cyclic components. The diagnostic summary is reported in Table 3, and implementation details are further described in Appendix A.3 on pages 30--31. All changes are marked in blue in the revised manuscript.
>
> **Q2: Implementation on ties**
>
> Thanks for the comment. In our graph representation, ties are handled directly in the cycle-counting step. A tie between two models is represented as two reciprocal directed edges, so tie-related structures are included when counting directed triangles and quadrilaterals. However, cycles formed purely by mutual ties should not be treated as harmful preference inconsistency. We therefore count pure-tie cycles separately and subtract them from the total short-cycle counts. The resulting bad-cycle count captures nontrivial preference contradictions while excluding cycles caused only by mutual ties.
>
> **Action:** We clarified how ties are represented and counted in the graph formulation and cycle-counting procedure. The main text now states that ties are represented by reciprocal directed edges in Section 2.1 on page 6, and Section 3.1 on pages 8--9 explains how pure-tie cycles are subtracted from the bad-cycle count. All changes are marked in blue in the revised manuscript.

---

> > ### Comment · Reviewer_EG8t · 2026-06-21
> >
> > Thank you for your response! I really appreciate the detailed explanations and the additional experiments, which have successfully addressed most of my questions. However, after reviewing the updated manuscript, I still have some reservations regarding the intrinsic evaluation of the resulting model rankings. Specifically, it would be highly beneficial to explicitly describe the experimental settings and clarify how the goodness-of-fit test was applied, as I could not find these specific details in the cited paper (Jiang et al., 2012).

---

> > > ### Author Response · Authors · 2026-06-23
> > > **Response to reviewer EG8t**
> > >
> > > Thank you for pointing this out. We have revised the text before Table 3 to make the goodness-of-fit diagnostic self-contained. Specifically, we now define the diagnostic quantities used in the table: normalized deviance, maximum standardized pairwise residual, and Hodge-style cyclic residual. We also clarify that the connection to Theorems 2 and 3 in Jiang et al. (2011) is through the Hodge-style cyclic residual, which measures the part of the comparison flow that cannot be explained by any global score vector. We further explicitly describe the experimental setting: the diagnostic is applied separately for each judge model, selection rule, and MT-Bench category, using the retained comparison graphs corresponding to that setting.
> > >
> > > **Action:** We added the definitions and described the experimental setting before Table 3 on pages 15. All new changes are marked in blue.

---

> ### Author Response · Authors · 2026-06-12
> **Response to reviewer EG8t (part 2)**
>
> **Q3: Additional benchmark**
>
> Thanks for the suggestion. We agree that evaluating on additional benchmarks can better support the generality of the proposed method. We add supplementary experiments on HumanEval and MATH, where we apply the same task-prompt perturbation and cycle-aware filtering pipeline beyond the main MT-Bench setting. The results are shown below; lower distance is better.
>
> *Additional HumanEval and MATH experiments. Values are normalized Spearman distances reported in units of $10^{-3}$.*
>
> | Method | HumanEval / Coding | MATH / Math |
> | --- | --- | --- |
> | Trunc50→40 | **82 ± 1** | **13 ± 1** |
> | Block top-2 | 95 ± 0 | 23 ± 0 |
> | Block top-1 | 100 ± 0 | 31 ± 0 |
> | NoTrunc boot | 92 ± 2 | 21 ± 2 |
> | ScoreWin80 | 93 ± 1 | 20 ± 1 |
> | Random50→40×10 | 92 ± 1 | 24 ± 1 |
> | SingleGraph1 | 96 ± 0 | 18 ± 0 |
>
> Overall, the proposed truncation method performs best on both HumanEval and MATH. In particular, Trunc50→40 achieves the lowest distance on both tasks.
>
> **Action:** We added supplementary experiments on HumanEval and MATH in Appendix C.2, with the results reported in Table 7 on pages 43--44. All changes are marked in blue in the revised manuscript.
>
> **Q4: Ablation studies**
>
> Thanks for the suggestion. We agree that sorting all perturbed prompts globally may select multiple variants from the same original prompt, so it is useful to compare with a block-wise selection rule.
>
> We added the suggested block-wise ablation. Specifically, for each original prompt, we sort its $(m+1)$ prompt variants by the cycle-based score and retain the best variants within each block. We report two versions, BlockTop1 and BlockTop2, which keep the best one or best two variants from each original-prompt block. Overall, the original Trunc25→20 method still achieves the best average performance under both judge models, while the block-wise variants are competitive on some individual categories. This suggests that global cycle-based truncation is still effective, and the gain is not solely due to selecting semantically redundant prompt variants.
>
> **Action:** We added block-wise ablation variants, BlockTop1 and BlockTop2, to the main baseline comparison in Table 2 on pages 12--13. We also described the corresponding block-wise selection rule in Appendix A.3 on page 29. All changes are marked in blue in the revised manuscript.
>
> **Q5: Verification on perturbed prompts**
>
> Thanks for the suggestion. In fact, we have already included this check in the manuscript. In the Appendix C on the "Semantic equivalence checking prompt", we use an additional prompt to verify whether each perturbed prompt is semantically equivalent to the original one.
>
> **Action:** We made the semantic-equivalence checking step more explicit in the main text on page 10 and included the full semantic-equivalence checking prompt in Appendix C.1 on page 38. All changes are marked in blue in the revised manuscript.
>
> **Q6: Perturbation of the LLM judge**
>
> Thanks for the suggestion. We added an experiment on the coding and math categories. In this experiment, instead of constructing five comparison graphs from prompt perturbations, we construct five comparison graphs from judge-prompt perturbations and then apply the same cycle-aware truncation pipeline.
>
> The results are shown below. Lower distance is better. The proposed truncation method still achieves the best performance under judge-prompt perturbation on both coding and math.
>
> *Judge-prompt perturbation experiment on coding and math. Lower distance is better.*
>
> | Method | Coding | Math |
> | --- | --- | --- |
> | Trunc25→20 | **57 ± 2** | **31 ± 1** |
> | BlockTop2 | 74 ± 0 | 32 ± 0 |
> | BlockTop1 | 70 ± 0 | 38 ± 0 |
> | NoTrunc boot | 66 ± 2 | 43 ± 2 |
> | ScoreWin40 | 65 ± 2 | 37 ± 1 |
> | Random50→20×10 | 66 ± 1 | 44 ± 1 |
> | SingleGraph1/all10 | 60 ± 0 | 44 ± 0 |
>
> These results suggest that our graph-level cycle filtering can also be applied when the comparison graphs are generated by perturbing the judge prompt rather than the evaluated prompt.
>
> **Action:** We added a judge-prompt perturbation experiment in Appendix C.2, with results reported in Table 7 on pages 43--44. We also described how judge-prompt variants are encoded and passed through the same graph-selection pipeline in Appendix A.3 on page 31. All changes are marked in blue in the revised manuscript.

---

### Review · Reviewer_w2pn · 2026-05-29

**Summary Of Contributions:**

This paper studies the problem of pairwise LLM evaluation. The main concern is that pairwise comparisons may be intransitive, so the induced comparison graph may contain cycles and may not support a stable global ranking. To address this issue, the authors propose a prompt perturbation framework. For each prompt, they generate several semantically similar perturbed prompts, construct comparison graphs from the resulting pairwise judgments, and filter out graphs with many short cycles. The remaining comparison results are then aggregated using standard ranking models such as the Davidson model.

The paper also provides experiments on MT-Bench with 20 target LLMs and two judge models. The main empirical result is that the proposed truncation method improves the average agreement with an Arena-derived reference ranking compared with several baselines. The paper further provides a theoretical analysis under a stylized random directed graph model, showing that cycle-based truncation can reduce the number of graphs needed for exact recovery under certain assumptions.

Overall, the paper addresses an important problem in LLM evaluation. The idea of using graph-level structural consistency before ranking aggregation is natural and potentially useful.

**Audience:**

Yes

**Broader Impact Concerns:**

The paper studies LLM evaluation methodology, so the direct ethical risks are limited. However, the proposed method may still affect how LLM leaderboards are constructed and interpreted. If cycle-based filtering removes difficult, ambiguous, or controversial prompts, the resulting ranking may overrepresent easier cases and give a misleading impression of model reliability.

Another concern is that the method relies on LLM-as-a-judge outputs. If **the judge model has systematic biases**, the proposed filtering procedure may produce a ranking that is internally consistent but still biased. In particular, a low cycle count only indicates structural consistency; it does not guarantee agreement with human preferences, fairness across task domains, or fairness across model families.

I recommend that the authors discuss these limitations more explicitly. They should clarify that improved graph consistency does not automatically imply improved evaluation accuracy, human alignment, or fairness.

**Claims And Evidence:**

Yes

**Requested Changes:**

The following changes would further strengthen the work:

- First, the authors should clarify **what happens when the Bradley--Terry or Davidson model is misspecified**. The proposed method ultimately produces a score-based global ranking, but real LLM preferences may be task-dependent or genuinely non-transitive. If the true pairwise preference structure cannot be represented by a latent score model, it is unclear what the final ranking means. The authors should discuss the target estimand under model misspecification and whether cycle truncation still improves the relevant error.

- Second, the authors should **discuss or extend the theory beyond the current total-order assumption**. Section 5.1 assumes that the ground-truth comparison graph is induced by a chain, $1 \succ 2 \succ \cdots \succ n$. Under this assumption, all cycles are treated as noise. However, the motivation of the paper is that real LLM evaluations may contain intransitivity. If the latent comparison graph itself contains cycles, Theorem 1 does not directly apply. The authors should explain whether an additional irreducible inconsistency term would appear in this setting.

- Third, the authors should **provide a more principled justification for the keep-$K$ parameter**. The current choice appears to be mainly empirical. A data-driven rule, such as ranking stability, held-out pairwise prediction, an elbow criterion, or validation against a small set of human judgments, would make the method more convincing and easier to use.

- Fourth, the authors should provide stronger evidence or further explanation (or discussion) that **low-cycle graphs are more accurate, not only more internally consistent**. A graph can be transitive but still wrong if the judge model is systematically biased. The authors should test whether retained graphs have higher agreement with human judgments or better out-of-sample predictive performance.

- Fifth, the authors should **add equal-budget baselines**. Since prompt perturbation increases the number of judge calls, the comparison should control for the total evaluation budget. Otherwise, it is difficult to know whether the improvement comes from cycle-aware filtering or simply from using more comparisons.

**Strengths And Weaknesses:**

# Strengths

- The problem is important and timely. Pairwise LLM evaluation is widely used, and cyclic or inconsistent preferences can make leaderboards unstable and hard to interpret.

- The proposed method is simple and modular. It can be combined with existing ranking methods such as Bradley--Terry models, and the graph-filtering step is easy to understand.

- The paper provides both empirical and theoretical results. The experiments compare the proposed method with several baselines, and the theory gives some intuition for why removing inconsistent graphs may help.

- The empirical results are promising in average performance. The proposed truncation method achieves the best macro-average ranking distance under both GPT-5 and Prometheus judges.

# Weaknesses

- My main concern is that **the paper still relies on Bradley--Terry or Davidson-type ranking models after filtering**. These models assume that pairwise preferences can be explained by latent scores. However, in LLM evaluation, preferences may be task-dependent or genuinely non-transitive. If the Bradley--Terry or Davidson model is misspecified, it is unclear what the final ranking means. In such cases, cycle filtering may remove valid preference information rather than only removing noise.

- **The theoretical model is too idealized**. In Section 5.1, the ground-truth comparison graph is assumed to come from a single total order. Therefore, all cycles are treated as noise. But the motivation of the paper is that real LLM comparisons may contain intransitivity. If the true comparison structure itself contains cycles, Theorem 1 no longer directly applies. In that case, there should be an additional irreducible inconsistency term, and the paper does not analyze this setting.

- **The choice of the keep-$K$ parameter is not well justified**. The paper shows empirically that an intermediate value of $K$ works well, but there is no theoretical or principled data-driven rule for choosing $K$. It is unclear how $K$ should depend on the number of models, the number of prompts, the perturbation budget, or the judge reliability.

- Another concern is that **low cycle count does not necessarily imply high accuracy**. A comparison graph can be very consistent but still wrong, especially if the judge model has systematic bias. Therefore, filtering graphs with fewer cycles may improve internal consistency, but it does not necessarily mean the retained graphs are closer to human preference.

- The comparison with **baselines does not fully isolate the source of improvement**. Prompt perturbation increases the number of judge calls, and the current results do not clearly show whether the gain comes from cycle-aware filtering, from using more comparisons, or from the perturbation procedure itself. Equal-budget baselines would be needed for a cleaner comparison.

---

> ### Author Response · Authors · 2026-06-12
> **Response to reviewer w2pn (part 1)**
>
> We sincerely thank the reviewer for the valuable comments. We address the reviewer's questions below and have also run several additional analyses to clarify the role of cycle-based filtering, the choice of the keep-$K$ parameter, and the equal-budget comparison.
>
> **Q1. Potential misspecification on ranking models**
>
> Thanks for the important question. We agree that Bradley--Terry and Davidson models are score-based aggregation models, and that a single score vector may not fully capture task-dependent or genuinely non-transitive LLM preferences. Our empirical analysis is therefore not intended to imply that all tasks share one universal preference structure. In fact, as shown in Table 2, we report results separately across the eight MT-Bench domains, and the behavior of different methods varies across domains.
>
> To further examine whether our conclusions depend on the Bradley--Terry specification, we added a diagnostic analysis for possible Bradley--Terry misspecification. The diagnostics evaluate whether the fitted Bradley--Terry scores adequately explain the observed pairwise comparisons, using residual errors, cyclic components, and held-out prediction as evidence. As shown in Table 2 in the manuscript, after cycle-based truncation there are no likely misspecification cases for either judge model. For GPT-5, likely misspecification cases decrease from 2 under no truncation to 0 after truncation; for both judges, block top-$K$ also increases the number of no-strong-warning cases compared with using all comparisons. These results suggest that the retained comparison graphs are more compatible with Bradley--Terry-style score-based aggregation.
>
> | Judge | Selection | Likely | Possible | No strong | $\Delta_{\mathrm{cv}}$ |
> | --- | --- | --- | --- | --- | --- |
> | GPT-5 | block top-$K$ | 0 | 2 | 6 | -0.007 |
> | GPT-5 | global top-$K$ | 0 | 3 | 5 | -0.007 |
> | GPT-5 | no truncation | 2 | 2 | 4 | -0.001 |
> | Prometheus | block top-$K$ | 0 | 2 | 6 | -0.005 |
> | Prometheus | global top-$K$ | 0 | 4 | 4 | -0.004 |
> | Prometheus | no truncation | 0 | 3 | 5 | -0.003 |
>
> At the same time, we agree that misspecification remains a real limitation. If some cycles reflect stable task-dependent preferences rather than judge noise, cycle filtering may remove valid preference information, especially in open-ended domains such as writing where different criteria or styles can lead to different preferences. We therefore do not claim that every discarded cycle is invalid. In our experiments, however, we did not observe evidence of such systematic loss; instead, cycle truncation improves ranking performance across the considered judges and evaluation settings, with domain-wise results in Table 2 showing where the gains are stronger or weaker.
>
> **Action:** (1) We added Bradley--Terry misspecification diagnostics in Section 4.3, "Diagnostics and Sensitivity Analyses," with the diagnostic summary reported in Table 3 on pages 14--15. (2) We also clarified the task-dependent nature of the evaluation in Table 2 on pages 12--13 and added a limitation discussion on score-model misspecification in Section 6 on page 22. All changes are marked in blue in the revised manuscript.
>
> **Q2: Theoretical model**
>
> Thanks for the question. Theorem 1 does not require the ground-truth comparisons to be induced by a single total order. Instead, the proof treats each pairwise edge $(i,j)$ separately: we first control the error probability of estimating each edge via majority voting, and then apply a union bound over all pairs. Therefore, the argument does not rely on global transitivity. Even if the ground-truth comparison graph contains cycles, the comparison graph can still be recovered under the stated conditions.
> In such a setting, the recovered comparison graph would itself be cyclic, reflecting genuine non-transitive preference structure rather than estimation error. Thus, Theorem 1 should be understood as an edgewise recovery guarantee for the latent comparison graph, not as a guarantee that the latent preferences are globally transitive.
>
> We also clarify that this total order is especially important for Theorem 2. Theorem 2 conditions on the retained graph having no directed triangle. The support is restricted to transitive tournaments, equivalently the $n!$ possible total orders. Therefore, if the latent preference structure is genuinely non-transitive, Theorem 2 no longer recovers that cyclic structure exactly. Instead, an additional irreducible approximation term may remain, and the resulting ranking should be interpreted as the closest transitive approximation to the latent comparison relation under the chosen loss.
>
> **Action:** We revised Section 5.2 on pages 19--21 to clarify the roles of Theorems 1 and 2. All changes are marked in blue in the revised manuscript.

---

> ### Author Response · Authors · 2026-06-12
> **Response to reviewer w2pn (part 2)**
>
> **Q3: Justification on the choice of parameters**
>
> Thanks for the suggestion. To make the choice of the keep-$K$ parameter more principled, we added a data-driven selection rule based on cross-validation. Specifically, we use grouped cross-validation over original prompt families. For each candidate $K$, we fit the ranking model on the retained low-cycle graphs from the training folds and evaluate the held-out pairwise prediction loss on validation folds. We then select $K$ using a one-standard-error rule together with a bootstrap stability check. The selected values of $K$ are mostly in the middle of the admissible range, which is consistent with our original empirical observation and conclusion: choosing $K$ too small discards useful information, while choosing $K$ too large reintroduces noisy and inconsistent graphs.
>
> Other data-driven choices of $K$ are also possible. For example, when a small set of human judgments is available, $K$ can be selected or calibrated by validation against human agreement. We use grouped cross-validation because it requires no additional human annotation and is directly applicable to our setting.
>
> *Data-driven selection of the keep-$K$ parameter.*
>
> | Data | Category | Chosen $K$ |
> | --- | --- | --- |
> | GPT-5 | coding | 28 |
> | GPT-5 | extraction | 25 |
> | GPT-5 | humanities | 10 |
> | GPT-5 | math | 20 |
> | GPT-5 | reasoning | 30 |
> | GPT-5 | roleplay | 30 |
> | GPT-5 | stem | 20 |
> | GPT-5 | writing | 20 |
> | Prometheus | coding | 18 |
> | Prometheus | extraction | 40 |
> | Prometheus | humanities | 13 |
> | Prometheus | math | 13 |
> | Prometheus | reasoning | 18 |
> | Prometheus | roleplay | 13 |
> | Prometheus | stem | 28 |
> | Prometheus | writing | 15 |
>
> **Action:** (1) We added a data-driven keep-$K$ selection procedure based on grouped cross-validation in Section 4.3 on pages 16--17, with the selected budgets reported in Table 4. (2) We also added a brief discussion that other choices, such as validation against a small set of human judgments, are possible when such labels are available. All changes are marked in blue in the revised manuscript.
>
> **Q4: Relation between low cycle counts and high accuracy**
>
> We thank the reviewer for raising this important point. We agree that a low cycle count is not a theoretical sufficient condition for high ranking accuracy. To substantiate this connection, we added new experimental results in the revised manuscript. Specifically, we plot the cycle-based score against the distance to the gold-standard ranking and fit the corresponding trend curves. Across two different judge models, we observe a clear positive correlation: rankings with larger cycle scores tend to have larger distances from the gold standard, while rankings with fewer cycles are consistently closer to the gold standard. These results provide empirical evidence that cycle counts serve as an effective reliability indicator for LLM-based evaluation in our setting.
>
> Although this empirical relationship is encouraging, it does not mean that low cycle count is sufficient for accurate evaluation in general. For example, if a judge has a stable systematic bias, it may produce a comparison graph that is internally consistent but still misaligned with human preferences. We therefore interpret cycle count as an internal reliability signal, not as a standalone guarantee of evaluation accuracy or human alignment.
>
> **Action:** (1) We added an empirical analysis of the relation between cycle-based scores and ranking accuracy in Section 4.2, with Figure 2 on page 12. (2) We also added a limitation discussion clarifying that low cycle count is an internal reliability signal, not a standalone guarantee of evaluation accuracy or human alignment. All changes are marked in blue in the revised manuscript.

---

> ### Author Response · Authors · 2026-06-12
> **Response to reviewer w2pn (part 3)**
>
> **Q5: Identification on different source of improvement**
>
> Thanks for the suggestion. It is important to isolate the source of improvement. In our setting, since the sampling temperature is set to 0, obtaining more independent comparisons mainly requires using more prompt perturbations. Therefore, the improvement can be decomposed into two factors: using more perturbed comparisons and applying cycle-aware truncation.
>
> To further isolate the effect of truncation, we added a budget-sweep experiment. For each fixed graph budget, we compare the ranking distance with and without truncation; the difference is reported as the gain. As shown below, increasing the graph budget improves both methods, while truncation consistently gives lower distance. Moreover, the gain becomes larger when more graphs are available, showing that cycle-aware filtering becomes more effective when there are enough perturbation candidates to select from.
>
> *Budget-sweep comparison between no truncation and cycle-aware truncation. Lower distance is better; cleaning gain is computed as no truncate minus truncate.*
>
> | Judge | Graphs | No truncate | Truncate | Cleaning gain |
> | --- | --- | --- | --- | --- |
> | GPT-5 | 50 | 0.0685 | 0.0567 | 0.0118 |
> | GPT-5 | 40 | 0.0684 | 0.0600 | 0.0085 |
> | GPT-5 | 30 | 0.0713 | 0.0634 | 0.0078 |
> | GPT-5 | 20 | 0.0774 | 0.0721 | 0.0052 |
> | GPT-5 | 10 | 0.0924 | 0.0878 | 0.0046 |
> | Prometheus | 50 | 0.0832 | 0.0695 | 0.0138 |
> | Prometheus | 40 | 0.0870 | 0.0753 | 0.0116 |
> | Prometheus | 30 | 0.0866 | 0.0777 | 0.0089 |
> | Prometheus | 20 | 0.0973 | 0.0934 | 0.0038 |
> | Prometheus | 10 | 0.1135 | 0.1129 | 0.0006 |
>
> This result is consistent with the equal-budget comparison in the original table: comparing Trunc25 with NoTrunc boot reflects the effect of cycle-aware filtering, while comparing NoTrunc boot with Single reflects the effect of using perturbations and more comparisons. Overall, both factors improve ranking quality, and the additional gain from cycle-aware truncation becomes more pronounced as the perturbation budget increases.
>
> **Action:** (1) We added a cost-matched comparison between truncated and untruncated aggregation in Figure 3 on page 14. (2) We also clarified the budget-matched resampling rules for the baselines in Appendix A.3 on pages 29--30. All changes are marked in blue in the revised manuscript.
>
> **Q6: Systematic bias on the judge model**
>
> Thanks for pointing this out. We agree that LLM-as-a-judge can suffer from systematic biases, and that such biases should be checked separately from graph consistency. In general, systematic judge bias can be diagnosed through controlled tests, such as swapping the answer order to test position bias, changing response length or style while preserving content to test verbosity or style bias, stratifying residuals by domain or model family, checking tie rates under different tie-handling rules, or validating a small subset against human judgments.
>
> In our experiments, we explicitly implement one such diagnostic and mitigation step for position bias. As described in the "Judge prompt" paragraph of Appendix C.1, for each pair of responses we evaluate both orders, $(A,B)$ and $(B,A)$. If the two comparisons give consistent preferences after swapping the order, we keep the resulting pairwise decision. If they are inconsistent, we use the assigned scores to determine the final comparison outcome. This symmetric evaluation protocol reduces the influence of position bias, since each response appears once in each position. We have clarified this point in the main text and revised the manuscript to explicitly discuss this debiasing step.
>
> Finally, as also discussed in our response to Q4, we do not claim that low cycle count provides a general guarantee of evaluation accuracy, human alignment, or fairness. A judge with stable systematic bias may still produce an internally consistent comparison graph. Cycle count is an internal reliability signal, and in our experiments it is empirically associated with better agreement with the Arena-derived reference ranking.
>
> **Action:** (1) We clarified the position-debiased Prometheus protocol in Section 4.1 on page 11 and Appendix C.1 on page 39. (2) We also added discussion of broader systematic judge biases and emphasized that graph consistency does not by itself guarantee accuracy, human alignment, or fairness in Section 6 on page 23. All changes are marked in blue in the revised manuscript.

---

> ### Comment · Reviewer_w2pn · 2026-06-20
>
> Thanks for the responses. All the concerns have been addressed. I especially appreciate the authors' additional diagnostics on Bradley--Terry misspecification.
>
> My understanding is that the proposed truncation procedure essentially retains comparison graphs with fewer cyclic inconsistencies, making the retained data more compatible with BTL-style score-based aggregation. This interpretation is also supported by the authors' diagnostics: after cycle-based truncation, the number of likely Bradley--Terry misspecification cases decreases, suggesting that reducing cycles improves the fit of a score-based ranking model.
>
> I think this finding is closely related to the recent AoS paper *Statistical Impossibility and Possibility of Aligning LLMs with Human Preferences: From Condorcet Paradox to Nash Equilibrium* by Kaizhao Liu, Qi Long, Zhekun Shi, Weijie J. Su, and Jiancong Xiao. That paper shows that **the absence of Condorcet cycles is a sufficient and necessary condition under which scalar reward/score models can represent human preferences**. Although the settings are not identical, this theoretical result seems to provide useful support for the authors' empirical observation that cycle truncation improves compatibility with Bradley--Terry-style aggregation. This might also be related to Q4.
>
> I noticed that this paper is not currently cited in the manuscript. I would encourage the authors to briefly discuss it, since it provides a relevant theoretical perspective on why removing cyclic inconsistencies can make score-based ranking more reliable.

---

> > ### Author Response · Authors · 2026-06-23
> > **Response to reviewer w2pn**
> >
> > Thank you for pointing out this connection. Liu et al. (2025) provided a useful theoretical perspective for our work. We have added a discussion in the related-work and ranking-model sections, noting that the absence of Condorcet cycles is necessary and sufficient for scalar reward representation, which supports our interpretation that cyclic inconsistencies are problematic for score-based aggregation.
> >
> > We also added a short discussion in Section 5. After truncation to triangle-free tournaments, each retained graph becomes transitive and is represented by one of the $n!$ total orders. This is closely related to the linear-preference setting in Liu et al. (2025), where the preference support also consists of strict rankings over $n$ items.
> >
> > **Action:** We added these discussions on pages 3--4, 7, and 21 of the revised manuscript. All new changes are marked in blue.

---

### Decision · Action_Editor_9HXt · 2026-06-22

**Recommendation:** Accept with minor revision

**Comment:**

The authors have made great efforts in their revision. The main questions and concerns from the referees are well addressed. Please address the remaining minor concerns from the two reviewers before the paper can be accepted.

**Audience:**

Yes

**Claims And Evidence:**

Yes

---

> ### Author Response · Authors · 2026-07-14
> **Response to Action Editor 9HXt**
>
> Dear Action Editor 9HXt,
>
> We have addressed the remaining minor concerns in the camera-ready version and have also added the link to our code.
>
> Best regards,
>
> The Authors of “Prompt Perturbation for Reliable LLM Evaluation over Comparison Graphs”